# The protein phosphatase PPM1A dephosphorylates and activates YAP to govern mammalian intestinal and liver regeneration

**Ruyuan Zhou[1,2]☯, Qirou Wu[1]☯, Mengqiu Wang[1], Seema Irani[3], Xiao Li[1], Qian Zhang[1,2], Fansen Meng[1,2], Shengduo Liu[1], Fei Zhang[1], Liming Wu[2], Xia Lin[4], Xiaojian Wang[5], Jian Zou[6], Hai Song[1], Jun Qin[7], Tingbo Liang[2], Xin-Hua Feng[1,4], Yan Jessie Zhang[3], Pinglong Xu**[1,2] *

**1** MOE Laboratory of Biosystems Homeostasis & Protection, Zhejiang Provincial Key Laboratory for Cancer Molecular Cell Biology, Life Sciences Institute, Zhejiang University, Hangzhou, China, **2** Department of Hepatobiliary and Pancreatic Surgery and Zhejiang Provincial Key Laboratory of Pancreatic Disease, The First Affiliated Hospital, Zhejiang University School of Medicine, Hangzhou, China, **3** Department of Molecular Biosciences; Institute for Cellular and Molecular Biology, University of Texas at Austin, Austin, Texas, United States of America, **4** Michael E. DeBakey Department of Surgery and Department of Molecular and Cellular Biology, Baylor College of Medicine, Houston, Texas, United States of America, **5** Institute of Immunology, Zhejiang University School of Medicine, Hangzhou, China, **6** Eye Center of the Second Affiliated Hospital School of Medicine, Institute of Translational Medicine, Zhejiang University, Hangzhou, China, **7** CAS Key Laboratory of Tissue Microenvironment and Tumor, Center for Excellence in Molecular Cell Science, Shanghai Institute of Nutrition and Health, Chinese Academy of Sciences, Shanghai, China

☯ These authors contributed equally to this work.
* xupl@zju.edu.cn

**Data Availability Statement:** The authors confirm that all data underlying the findings are fully available without restriction. All relevant data are within the paper and its Supporting Information

## Abstract

The Hippo-YAP pathway responds to diverse environmental cues to manage tissue homeostasis, organ regeneration, tumorigenesis, and immunity. However, how phosphatase(s) directly target Yes-associated protein (YAP) and determine its physiological activity are still inconclusive. Here, we utilized an unbiased phosphatome screening and identified protein phosphatase magnesium-dependent 1A (PPM1A/PP2Cα) as the bona fide and physiological YAP phosphatase. We found that PPM1A was associated with YAP/TAZ in both the cytoplasm and the nucleus to directly eliminate phospho-S127 on YAP, which conferring YAP the nuclear distribution and transcription potency. Accordingly, genetic ablation or depletion of PPM1A in cells, organoids, and mice elicited an enhanced YAP/TAZ cytoplasmic retention and resulted in the diminished cell proliferation, severe gut regeneration defects in colitis, and impeded liver regeneration upon injury. These regeneration defects in murine model were largely rescued via a genetic large tumor suppressor kinase 1 (LATS1) deficiency or the pharmacological inhibition of Hippo-YAP signaling. Therefore, we identify a physiological phosphatase of YAP/TAZ, describe its critical effects in YAP/TAZ cellular distribution, and demonstrate its physiological roles in mammalian organ regeneration.

files. The MS raw files have been deposited to the Mass Spectrometry Interactive Virtual Environment (MassIVE MSV000086694). FTP Download Link: http://massive.ucsd.edu/MSV000086694/.

**Funding:** This research was sponsored by the National Natural Science Foundation of China (www.nsfc.gov.cn) Distinguished Young Scholars Program (31725017 to P.X.) and the Projects (31830052 to P.X. and 81902915 to Q.Z.). The funders had no role in study design, data collection and analysis, decision to publish, or preparation of the manuscript.

**Competing interests:** The authors have declared that no competing interests exist.

**Abbreviations:** BME, β-mercaptoethanol; CNS, central nervous system; DAI, disease activity index; dKO, double knockout; Dox, doxycycline; dph, day post hepatectomy; DSS, dextran sulphate sodium; FBS, fetal bovine serum; GPCR, G protein-coupled receptor; gRNA, guide RNA; HR, homozygous recombination; IFN-I, type I interferon; IHC, immunohistochemistry; IP, intraperitoneal; KO, knockout; LATS1, large tumor suppressor kinase 1; MAVS, mitochondrial antiviral signaling protein; MEF, mouse embryonic fibroblast; MLB, Myc lysis buffer; MST1, mammalian sterile 20-like kinase 1; PEI, Polyethylenimine; phospho-YAP, phosphorylating forms of YAP; PPM1A, protein phosphatase magnesium-dependent 1A; SEM, standard error of mean; Ser/Thr, serine/threonine; siRNA, small interfering RNA; SPF, specific-pathogen-free; STING, stimulator of interferon genes; TAZ, transcriptional coactivator with PDZ-binding motif; TEAD, transcriptional enhanced associate domain; TGF-β, transforming growth factor beta; TTF, tail-tip fibroblast; WT, wild-type; YAP, Yes-associated protein.

# Introduction

Originally discovered in *Drosophila*, the Hippo pathway is highly conserved in evolution [1–4]. Four tumor suppressors, constituting the Hpo-Sav/MST-SAV complex and the Wts-Mats/LATS-MOB complex, sequentially govern the cellular localization, activity, and fate of signaling effectors Yes-associated protein/transcriptional coactivator with PDZ-binding motif (YAP/TAZ). In response to unfavorable growth conditions and mediated by the upstream kinases including MST, MAP4Ks, TAO, and AMPK, LATS kinases phosphorylate YAP and TAZ [5–14]. Large tumor suppressor kinase 1 and 2 (LATS1/2)-mediated phosphorylation retains YAP/TAZ in the cytoplasm for sequestration, ubiquitination, and degradation [15,16]. Otherwise, YAP/TAZ are localized in the nucleus to form a transcription complex and activate the transcriptional enhanced associate domain (TEAD) family transcription factors, hereby transcribing target genes to promote cell proliferation, migration, and survival [17]. Physiological and pathological functions of the Hippo pathway are gradually established, particularly in the development, homeostasis, and regeneration of organs including liver, heart, intestine, brain and central nervous system (CNS), lung, kidney etc. [18], and in the diseases of cancer, immune disorder, cardiovascular dysfunction [17,19].

Regulations of YAP/TAZ are complex and cross-talk with the Notch pathway [20], the transforming growth factor beta (TGF-β) pathway [21], the WNT pathway [22], G protein-coupled receptor (GPCR) signaling [7,23], and innate immune signaling [24,25]. How Hippo-YAP signaling integrates the intrinsic factors and how it cooperates with other signaling pathways to regulate a variety of physiological and pathological processes are one of the major researches focuses in the Hippo-YAP field, but still remain largely unanswered.

Reversible phosphorylation is one of the fundamental regulatory mechanisms to regulate a myriad of biological processes. It raises an intriguing question about what protein phosphatases are involved in regulation of Hippo-YAP signaling and how they act. Among 150 known protein phosphatases, three of them are shown to regulate the Hippo-YAP pathway, including the STRIPAK-PP2A complex that targets Hpo/MST kinases [26–28], PP1 that selectively dephosphorylates TAZ and LATS1 [29,30], and Pez/PTPN14 that interacts the WW domain of YAP to translocate YAP from the nucleus to the cytoplasm, independent of its phosphatase activity [31]. Among them, STRIPAK-PP2A functions in the development and immunity of *Drosophila* [25,27,31] and regulates mammalian epidermal proliferation [32], and PTPN14 is involved in the cancer progression [33,34]. However, it remains unknown as to phosphatase(s) by which it directly targets the key phosphorylation modification of YAP. The physiological significance of dephosphorylation regulation to the Hippo-YAP mechanism in mammals is also elusive. Therefore, identification of the bona fide phosphatase may serve as a key to understand the complex signaling and regulatory mechanisms of the Hippo-YAP pathway.

Here, we identified that protein phosphatase magnesium-dependent 1A (PPM1A/PP2Cα), a ubiquitously expressed PP2C phosphatase [35], was a direct and bona fide modifier of YAP. We revealed that PPM1A was physiologically critical to the Hippo pathway in the murine models of organ regeneration. PPM1A deficiency led to marked phenotypes, including the down-regulated YAP/TAZ levels in the nucleus, compromised cellular proliferation, and substantially attenuated regeneration of intestine and liver, which could be rescued by genetic or pharmacological inhibition of Hippo signaling. Therefore, these findings elucidate a critical role of PPM1A in mammalian organ regeneration and provide physiological evidences for the value of reverse phosphorylation in Hippo-YAP signaling.

## Results

### PPM1A facilitates nuclear localization and transcription potency of YAP/TAZ

To systematically evaluate the serine/threonine (Ser/Thr) phosphatase(s) in regulation of Hippo-YAP signaling, we analyzed a cDNA library of Ser/Thr phosphatome (consisting of 40 members of Ser/Thr protein phosphatases) to the transcription activity of YAP, under mammalian sterile 20-like kinase 1 (MST1)-stimulated activation of Hippo signaling. The reporter assays revealed that PPM1A/PP2Cα, as well as previously reported PP2A [26–28], substantially enhanced the transcriptional potency of YAP-TEADs that were suppressed by MST1 (Fig 1A and S1A Fig). Two other phosphatases, including PPM1B and PP1γ, also attenuated Hippo signaling at a lesser degree (Fig 1A). In addition to relieving the MST1-driven inhibition of YAP (Fig 1B), PPM1A also reversed the YAP suppression triggered by LATS1 (Fig 1C), and MAP4K1 (Fig 1D), another upstream kinase of the LATS kinases [9]. The PPM1A-induced regulation of Hippo signaling obviously needed the phosphatase enzymatic activity of PPM1A, as a phosphatase-dead PPM1A mutant (D239N) failed to potentiate YAP (Fig 1B–1D). Depending on its enzymatic activity, PPM1A similarly potentiated the transcription potency of TAZ (S1B Fig). These consistent observations suggest that PPM1A functions as a negative regulator of the Hippo pathway and facilitates the activities of YAP and TAZ.

To validate the effect of PPM1A under physiological stimulations, we generated HEK293 cells for inducible PPM1A expression by a doxycycline (Dox) system. As expected, inducible expression of PPM1A by Dox impeded the energy stress-induced YAP phosphorylation on the S127 residue (Fig 1E), a key YAP modification that determines YAP cytoplasmic retention [36]. Phospho-YAP (S127) was clearly attenuated by the active PPM1A (Fig 1F). Similarly, the MST1-induced degradation of TAZ, a key consequence upon TAZ phosphorylation, was blocked by PPM1A (Fig 1G). PPM1A induction was also accompanied with an increasing activity of the TEAD-responsive promoter (S1C Fig). Consistent with these observations, we detected an elevated mRNA expression of CTGF and CYR61, two well-defined target genes of the YAP/TAZ-TEAD transcription complex (S1D Fig) [37–39], in the presence of induced PPM1A (Fig 1H). These observations suggest that PPM1A eliminates the key phosphorylating modification(s) of YAP/TAZ, directly or indirectly.

The Hippo-YAP pathway critically depends on the cellular localization of YAP/TAZ to mediate the extra- and intracellular communications. We thus analyzed the effect of PPM1A on cellular distribution of endogenous YAP/TAZ. Serum deficiency resulted in a rapid and obvious cytoplasmic distribution of YAP/TAZ, as expected (Fig 1I). PPM1A induction markedly decreased the cytoplasmic subset of YAP/TAZ and facilitated their accumulations in the nucleus even under nutrient deficiency environment, as evidenced by immunofluorescence imaging (Fig 1I) and the nucleocytoplasmic fractions (Fig 1J). These data suggest that PPM1A governs the cellular translocation of YAP/TAZ, in resting state as well as in response to Hippo signaling activation.

### Loss-of-function of PPM1A enhances Hippo signaling and inactivates YAP/TAZ

To further validate the effects of PPM1A on Hippo signaling, we generated the PPM1A knockout (KO) HEK293 cells by a CRISPR-based strategy. Genetic ablation of PPM1A, as verified by the anti-PPM1A immunofluorescence imaging, resulted in an obvious increase in cytoplasmic distribution of endogenous YAP/TAZ (Fig 2A). We also observed substantially more exaggerated mobility shift of endogenous YAP when PPM1A was absent, as revealed by Phos-Tag electrophoresis, indicating an overall higher level of YAP/TAZ phosphorylation in cells with

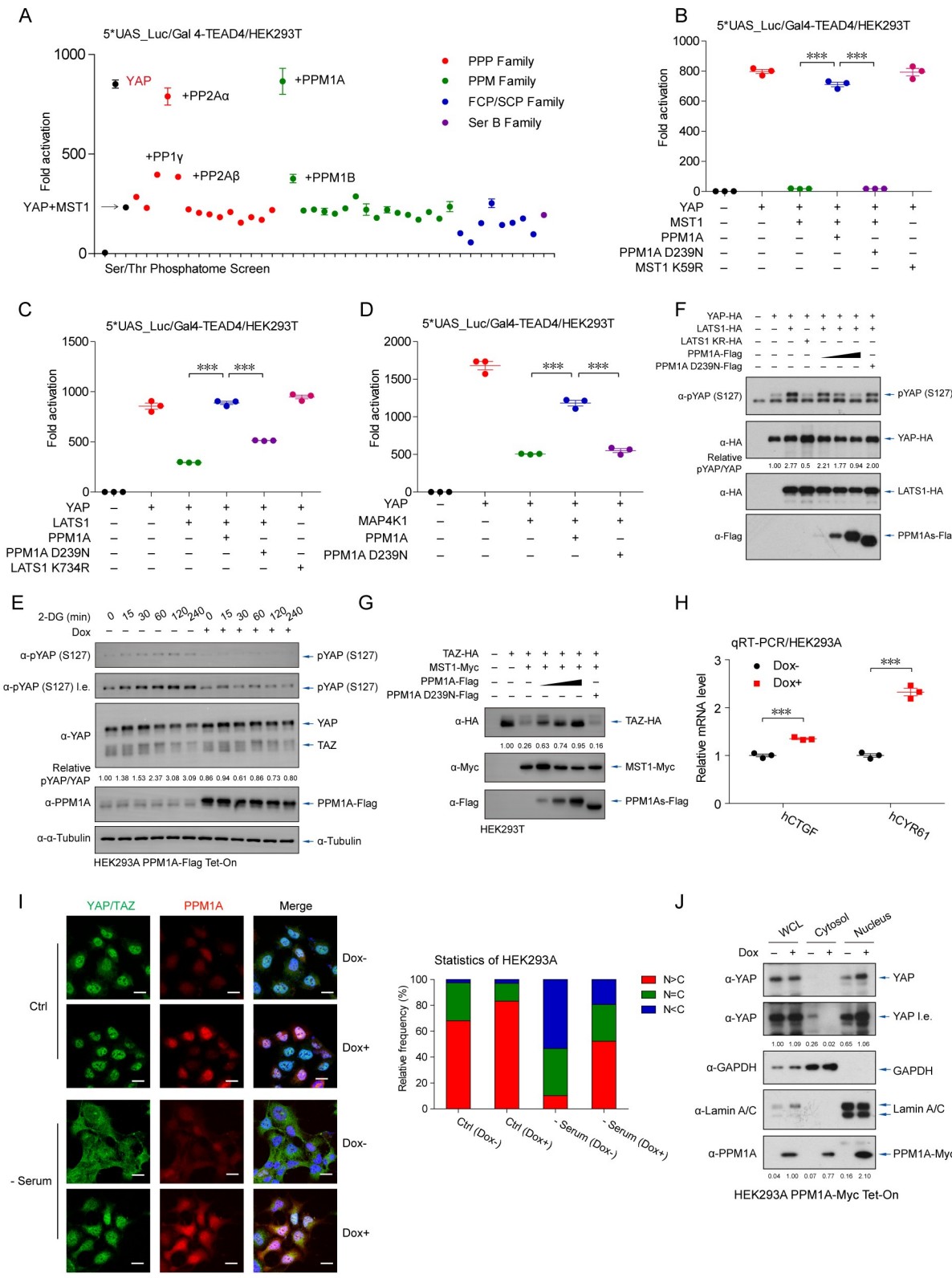

**Fig 1. PPM1A facilitates both nuclear distribution and transcription potency of YAP/TAZ. (A)** Transcription potency of YAP (5 ng), which drove the TEAD-responsive promoter, was substantially attenuated by coexpression of MST1 (10 ng). Using this setting, a cDNA library comprised of 40 members of serine/threonine phosphatase was screened for their individual contribution in regulating the Hippo-

YAP pathway. The phosphatome screen identified PPM1A/PP2Cα as a potent suppressor of Hippo-YAP signaling, which completely restored the MST1-induced suppression of the YAP-TEAD transactivation. (**B–D**) Transcription potency of YAP (5 ng), which was suppressed by coexpression of MST1 (50 ng) (B), LATS1 (500 ng) (C), or MAP4K1 (50 ng) (D), was markedly recovered by cotransfection of wild-type PPM1A but not the enzyme-dead PPM1A (D239N). (**E**) Phosphorylation of YAP at the S127 residue, which was revealed by immunoblotting and represented an activation status of Hippo signaling, was profoundly increased under 2-DG-triggered energy stress. Dox-induced PPM1A expression significantly attenuated this energy stress-triggered Hippo signaling. (**F**) The LATS1-trigged phospho-YAP (S127) was eliminated in a dose-dependent manner by coexpression of WT PPM1A, but not its phosphatase-dead form (D239N). (**G**) Coexpression of WT PPM1A stabilized TAZ in a dose-dependent manner, which otherwise was rapidly degraded upon activation of Hippo signaling. (**H**) The mRNA levels of CTGF and CYR61 were significantly up-regulated in the presence of PPM1A induction. (**I**) Serum starvation resulted in a substantial increase of YAP in the cytoplasm, which was significantly prevented by the induction of PPM1A (left panels). Percentage of cells with the relative ratio of nucleo-YAP vs cytoplasmic-YAP was counted (right panels). (**J**) Inducible PPM1A expression decreased the cytoplasmic-YAP but promoted nuclear accumulation of YAP, as analyzed by the nucleocytoplasmic fraction assay. Figs 1–7, unless otherwise indicated, $n = 3$ independent experiments (mean ± SEM). *, $P < 0.05$, **, $P < 0.01$, and ***, $P < 0.001$, compared with control condition (ANOVA test and Bonferroni correction). Unprocessed images of blots are shown in S1 Raw Images. Statistics source data are provided in S1 Data. Dox, doxycycline; LATS1, large tumor suppressor kinase 1; MST1, mammalian sterile 20-like kinase 1; PPM1A, protein phosphatase magnesium-dependent 1A; TAZ, transcriptional coactivator with PDZ-binding motif; TEAD, transcriptional enhanced associate domain; WT, wild-type; YAP, Yes-associated protein.

PPM1A deficiency (Fig 2B). PPM1A deficiency was accompanied with an enhanced level of phospho-S127 but the diminished TAZ protein level (Fig 2B and S2A and S2B Fig). Intriguingly, the degree of YAP phosphorylation in the resting state was comparable to those cells under glucose starvation (Fig 2B and S2A and S2B Fig), indicating a critical role of PPM1A in regulating YAP/TAZ activation. PPM1A deficiency also resulted in lower transactivation of TEAD-driven promoter (Fig 2C and S2C Fig) and down-regulation of CTGF and CYR61 mRNAs (Fig 2D and S2D and S2E Fig). These phenotypes were rescued by the reconstitution of PPM1A (Fig 2D and S2C Fig). Similarly, PPM1A reconstitution restored the level of phospho-YAP (S127) under PPM1A deletion and energy stress (Fig 2E). These consistent observations suggest that the level of endogenous PPM1A is a key determinant of phosphorylation, cellular distribution, and transcription potency of YAP/TAZ.

We next analyzed primary cells from the PPM1A KO mice, which were generated via a homogenous recombination strategy [40]. The striking increase of the cytoplasmic YAP/TAZ was detected in murine embryonic fibroblasts (MEFs) from the PPM1A KO homozygotes (Fig 2F and S2F Fig). Meanwhile, we observed a clearly enhanced level of YAP phospho-S112 (equivalent human S127) in the lysates of liver obtained from young PPM1A KO mice (S2G Fig), and the substantially lower levels of CTGF and CYR61 mRNAs in primarily cultured tail-tip fibroblasts (TTFs) from the PPM1A KO mice (Fig 2G). These data suggest that PPM1A is an intrinsic negative regulator of the Hippo-YAP pathway in mice.

We also employed small interfering RNAs (siRNAs) to deplete the endogenous expression of PPM1A in HEK293, human adenocarcinoma alveolar basal epithelial A549, and immortalized human normal keratinocyte HaCaT cells. The enhanced Hippo signaling, as evidenced by the increasing phospho-YAP (S127) and decreasing TAZ proteins, was observed upon depletion of PPM1A in HEK293 (Fig 2H), HaCaT (Fig 2I), and A549 cells (S2H Fig), both in resting state and under nutrient stresses. The compromised responsiveness of TEAD-driven promoter was also observed upon PPM1A depletion (Fig 2J), as well as lower expression of their target genes (Fig 2K and 2L). The mRNA levels of YAP/TAZ target genes in PPM1A-depleted cells appeared comparable to those cells under nutrient-stress conditions (Fig 2L). Additionally, reconstitution of PPM1A restored the nuclear distribution of endogenous YAP/TAZ in PPM1A KO cells (S2I Fig). These data support a critical role of PPM1A in regulating Hippo-YAP signaling and suggest that this PPM1A-mediated regulation is ubiquitous among cells with distinct origins.

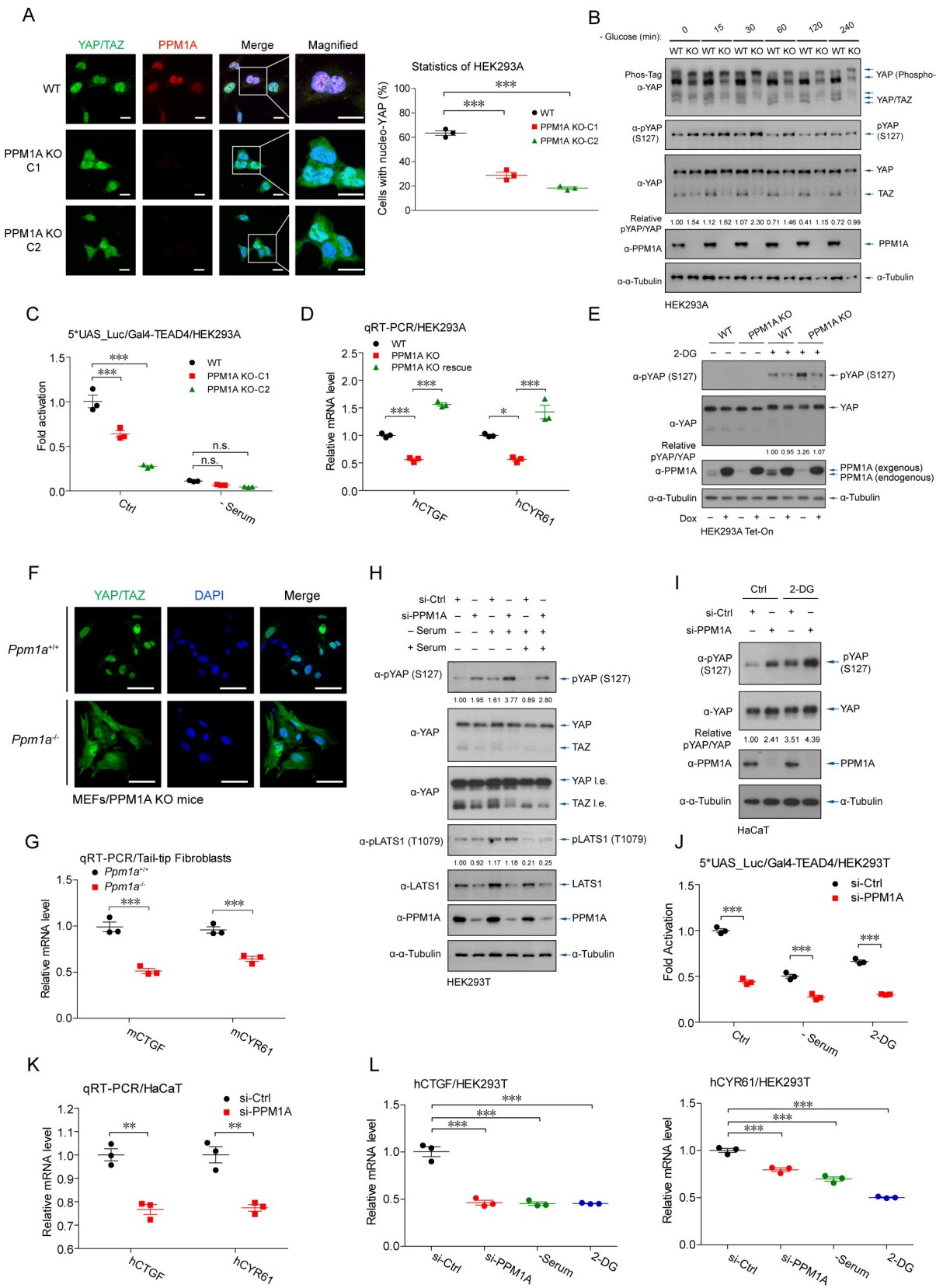

**Fig 2. Deletion or depletion of PPM1A enhances Hippo signaling and inactivates YAP/TAZ. (A**, **B)** PPM1A KO HEK293 cells were generated by CRISPR-based genome editing and verified by immunofluorescence (A) and immunoblotting (B). Genetic ablation of PPM1A resulted in a substantial increase of cytoplasmic YAP/TAZ (A), an up-regulation of phospho-YAP (S127) (B, second panel), a marked accumulation of highly phosphorylated YAP in a Phos-Tag electrophoresis (B, first panel), and a decrease of TAZ proteins (B, third panel). Glucose deficiency resulted in a phosphorylation status of YAP/TAZ similar to a degree of those cells with PPM1A deletion, as evidenced by the assays of mobility shift and phospho-YAP (S127) (B). **(C)** A compromised responsiveness to the TEAD-driven promoter was detected in cells without PPM1A. **(D)** Reconstitution of PPM1A in PPM1A KO cells via a viral-based delivery entirely recovered the expression of YAP/TAZ target genes. **(E)** Reintroduction of ectopic PPM1A in PPM1A KO cells alleviated the 2-DG-induced phosphorylation of endogenous YAP. The expression of endogenous and ectopic PPM1A was indicated by an anti-PPM1A immunoblotting. **(F)** Immunofluorescence imaging showed a profound increase of cytoplasmic YAP/TAZ in MEFs isolated from PPM1A KO mice, which were generated by a conventional HR strategy. **(G)** qRT-PCR assays for the TTFs of PPM1A KO mice showed a significant decrease of CTGF and CYR61 mRNAs. **(H–K)** Depletion of PPM1A in HEK293 cells (H) and HaCaT cells (I) by siRNA interferences resulted in an increased level of phospho-YAP (S127), either in resting state or states with nutrient deficiency. PPM1A depletion also down-regulated the transactivation of the YAP/TAZ-TEAD complex (J) and mRNAs of their target genes (K). **(L)** The levels of CTGF and CYR61 mRNAs in PPM1A-depleted cells were comparable to those cells under nutrient deficiency. Unprocessed images of blots are shown in S1 Raw Images. Statistics source data are provided in S1 Data. HR, homozygous recombination; KO, knockout; MEF, mouse embryonic fibroblast; PPM1A, protein phosphatase magnesium-dependent 1A; qRT-PCR, real-time quantitative PCR; siRNA, small interfering RNA; TAZ, transcriptional coactivator with PDZ-binding motif; TEAD, transcriptional enhanced associate domain; TTF, tail-tip fibroblast; YAP, Yes-associated protein.

## PPM1A translocates and associates with YAP/TAZ and LATS kinases

To dissect the precise mechanism underlying the PPM1A-mediated augment of YAP/TAZ activity, we first investigated the effects of PPM1A in MST1/2 and LATS1/2 double knockout (dKO) cells [41]. Depletion of PPM1A significantly decreased the mRNA expression of YAP/TAZ target genes in both wild-type and MST1/2 dKO cells, but not in LATS1/2 dKO cells (Fig 3A). Intriguingly, PPM1A was able to reverse the YAP transcriptional potential that was stimulated by a phosphomimetic LATS1 mutant (T1079D) [42] (Fig 3B and 3C), suggesting that PPM1A may directly target YAP/TAZ.

Analyzing the cellular distribution of PPM1A by immunofluorescence imaging revealed that PPM1A was mainly in the nucleus (Fig 3D). The cytoplasmic PPM1A, which only accounted for a small subset of total PPM1A, overlapped with YAP in the cytoplasm (Fig 3D). Unexpectedly, LATS1 kinase induced a marked cytosolic distribution of PPM1A, which partially overlapped with LATS1 (Fig 3D and 3E). Notably, in response to nutrient or energy stress, PPM1A was partially exported to the cytoplasm and associated with the similarly exported YAP, where they were well overlapped (Fig 3F). Furthermore, we detected the modest interactions between PPM1A and YAP/TAZ (Fig 3G), the endogenous complex of PPM1A-YAP in regenerating livers (Fig 3H), as well as the association between endogenous YAP and stably expressed PPM1A (Fig 3I) and the endogenous PPM1A-LATS1 complex (Fig 3J). These observations suggest that PPM1A forms the complex with both YAP/TAZ and LATS1 kinases. The underlying mechanism for LATS1-induced cellular translocation of PPM1A is intriguing and warrants for future investigation.

## PPM1A directly but selectively eliminates phosphorylation of YAP on several residues

We next examined whether PPM1A directly modifies YAP. For this purpose, YAP was coexpressed together with MST1 and SAV1 and purified using HA-tag, which obtained the phosphorylating forms of YAP (phospho-YAP). Phospho-YAP was subsequently incubated with the separately isolated PPM1A during the in vitro phosphatase assay. Apparently, the isolated PPM1A was active (S3A and S3B Fig) [40], and it directly eliminated phospho-YAP at the S127 residue (Fig 4A and S3B Fig). To confirm this result, we also purified recombinant PPM1A from the *E. coli*, and it similarly eliminated the phospho-YAP (S127), depending on its phosphatase enzymatic activity at the presence of $Mg^{2+}/Mn^{2+}$ (Fig 4B). Using the malachite green assay, we also quantified the phosphatase activity of PPM1A against phosphoryl peptide

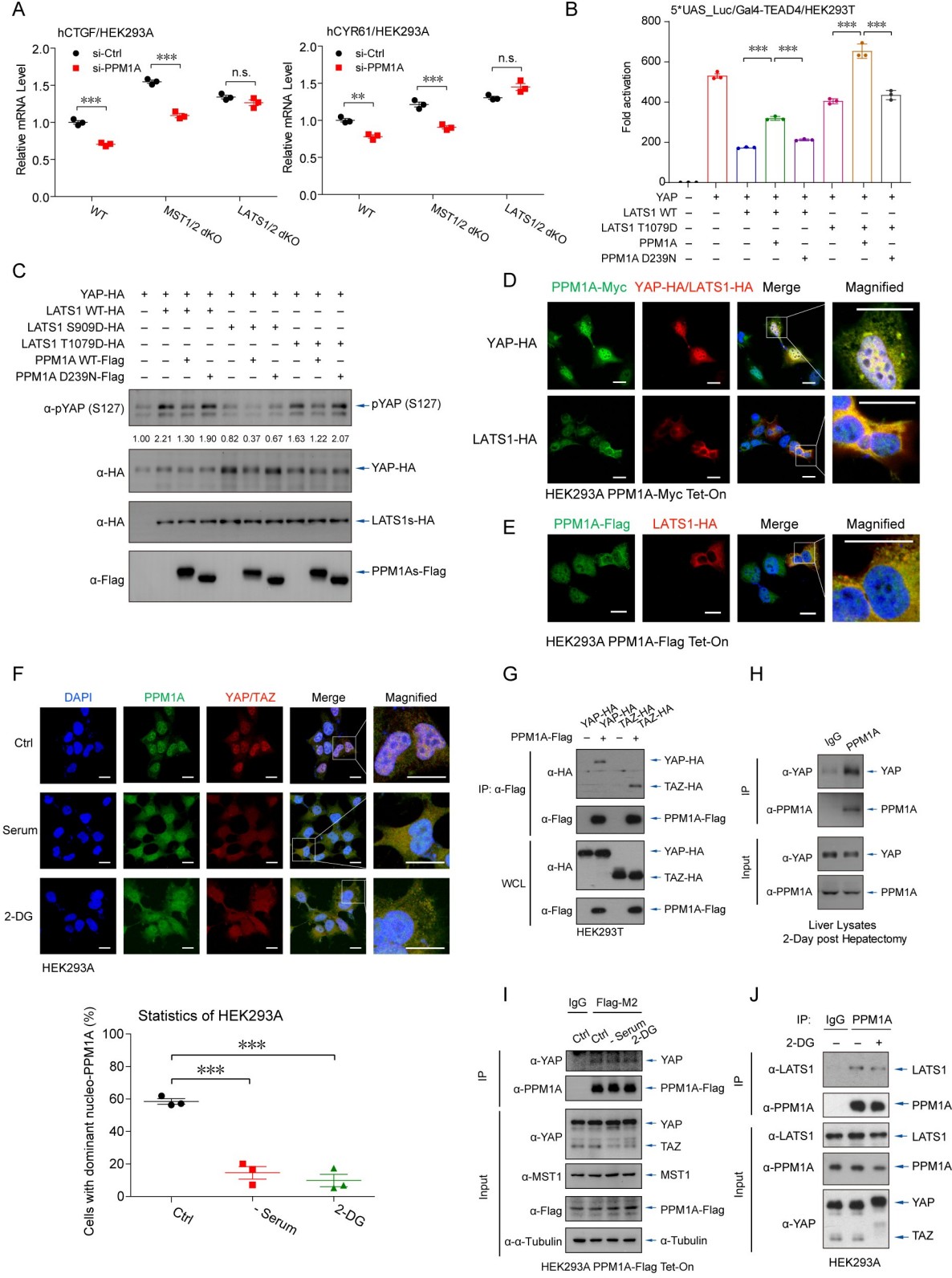

**Fig 3. PPM1A translocates and associates with YAP/TAZ and LATS kinases. (A)** Upon PPM1A depletion, qRT-PCR assays detected the significantly lower mRNA levels of YAP/TAZ target genes in WT or MST1/2 dKO cells, but not in LATS1/2 dKO cells. **(B, C)** The transcription potency of YAP that was inhibited by LATS1 T1079D, a LATS1 phosphomimetic, was restored by PPM1A cotransfection (B).

LATS1 T1079D-stimulated phospho-YAP (S127) was eliminated by PPM1A, similar to those phospho-YAP (S127) induced by WT LATS1 (C). LATS1 S909D failed to phosphorylate and inactivate YAP. **(D, E)**, Immunofluorescence imaging detected an overlap of cellular localization between PPM1A and YAP, occurred in the nucleus and in the cytoplasm (D). In the presence of LATS1, PPM1A translocated from the nucleus to the cytoplasm, where it overlapped with LATS1 (D and E). **(F)** In response to nutrient deficiency that activated Hippo signaling, endogenous PPM1A was partially exported to the cytoplasm, where it colocalized with YAP. Percentage of cells with above 50% PPM1A in the nucleus was counted (bottom panels). **(G–I)** The complex between PPM1A and YAP/TAZ was detected by coimmunoprecipitation assays, when these proteins were coexpressed (G), the endogenous YAP and PPM1A in liver lysates upon 2 dph (H), or between endogenous YAP and stably expressed PPM1A (I). **(J)** The endogenous complex of PPM1A-LATS1 was also visible in HEK293 cells, as revealed by coimmunoprecipitation assays using an anti-PPM1A antibody. Unprocessed images of blots are shown in S1 Raw Images. Statistics source data are provided in S1 Data. dKO, double knockout; dph, days post hepatectomy; LATS, large tumor suppressor kinase; PPM1A, protein phosphatase magnesium-dependent 1A; qRT-PCR, real-time quantitative PCR; TAZ, transcriptional coactivator with PDZ-binding motif; WT, wild-type; YAP, Yes-associated protein.

derived from YAP (sequence of QHVRAH-pSer-SPASLQ) with a specificity constant of $K_{cat}/K_m = 0.68 \pm 0.02$ mM$^{-1}$S$^{-1}$ (Fig 4C). Furthermore, using the Phos-Tag electrophoresis, we visualized that PPM1A eliminated a variety of but not all the MST1-induced YAP phosphorylation (Fig 4D). The mass spectrometry analysis to the in vitro phosphatase assay products revealed a subset of phospho-residues on YAP was selectively dephosphorylated by PPM1A, including the S109 and S366 (Fig 4E). We then constructed the YAP 2SA mutant (S109A/S127A) and reconstituted it into YAP-depleted gut epithelial cells (S3C Fig). Notably, this PPM1A-resistant YAP exhibited a higher potency to drive the transcription of YAP/TAZ-TEADs targets, and its activity was unaffected by PPM1A deficiency (Fig 4F). These data suggest that PPM1A directly targets and eliminates YAP phosphorylation on some of key residues, which facilitating its activation.

## PPM1A also targets the PRP4K-mediated YAP phosphorylation in the nucleus

The nuclear-localized kinase PRP4K is known to phosphorylate YAP on a subset of residues including S127 to restrict YAP nuclear accumulation and target gene transcription [43]. Given the factors that PPM1A directly targets YAP and PPM1A is distributed abundantly in the nucleus in many cells, we speculated that PPM1A might function as a phosphatase to safeguard YAP in the nucleus. Same as previously reported, expression of PRP4K led to a robust inhibition of YAP activity (Fig 4G), similar to but independent of LATS1/2 kinases (Fig 4H). Expression of wild-type PPM1A, but not the enzyme-dead mutant, reverted this PRP4K-imposed suppression (Fig 4G and 4H). Immunofluorescence imaging also revealed that PRP4K, which resided exclusively in the nucleus, induced an obvious increase of the cytoplasmic YAP/TAZ (Fig 4I and S3D Fig). In contrast, Dox-induced PPM1A expression abolished this PRP4K-mediated nuclear export of YAP/TAZ (Fig 4I and S3D Fig). These data suggest that PPM1A serves as the YAP phosphatase in the nucleus, and furthermore, validate again that PPM1A is a phosphatase targeting YAP.

## PPM1A is critical for proliferation of intestinal organoids

Because *Ppm1a*$^{-/-}$ mice display a largely normal phenotype in the development [40], we speculate that PPM1B, the phosphatase considerably related to PPM1A and capable also for Hippo signaling regulation (Fig 1A), may compensate for the function of PPM1A in Hippo regulation during the development. In agreement with this speculation, double depletion of PPM1A and PPM1B resulted in stronger YAP phosphorylation (Fig 5A) and substantially reduced colony formation (Fig 5B), and the offspring of *Ppm1a*$^{-/-}$ and *Ppm1b*$^{d/d}$ mice [44] is unobtainable.

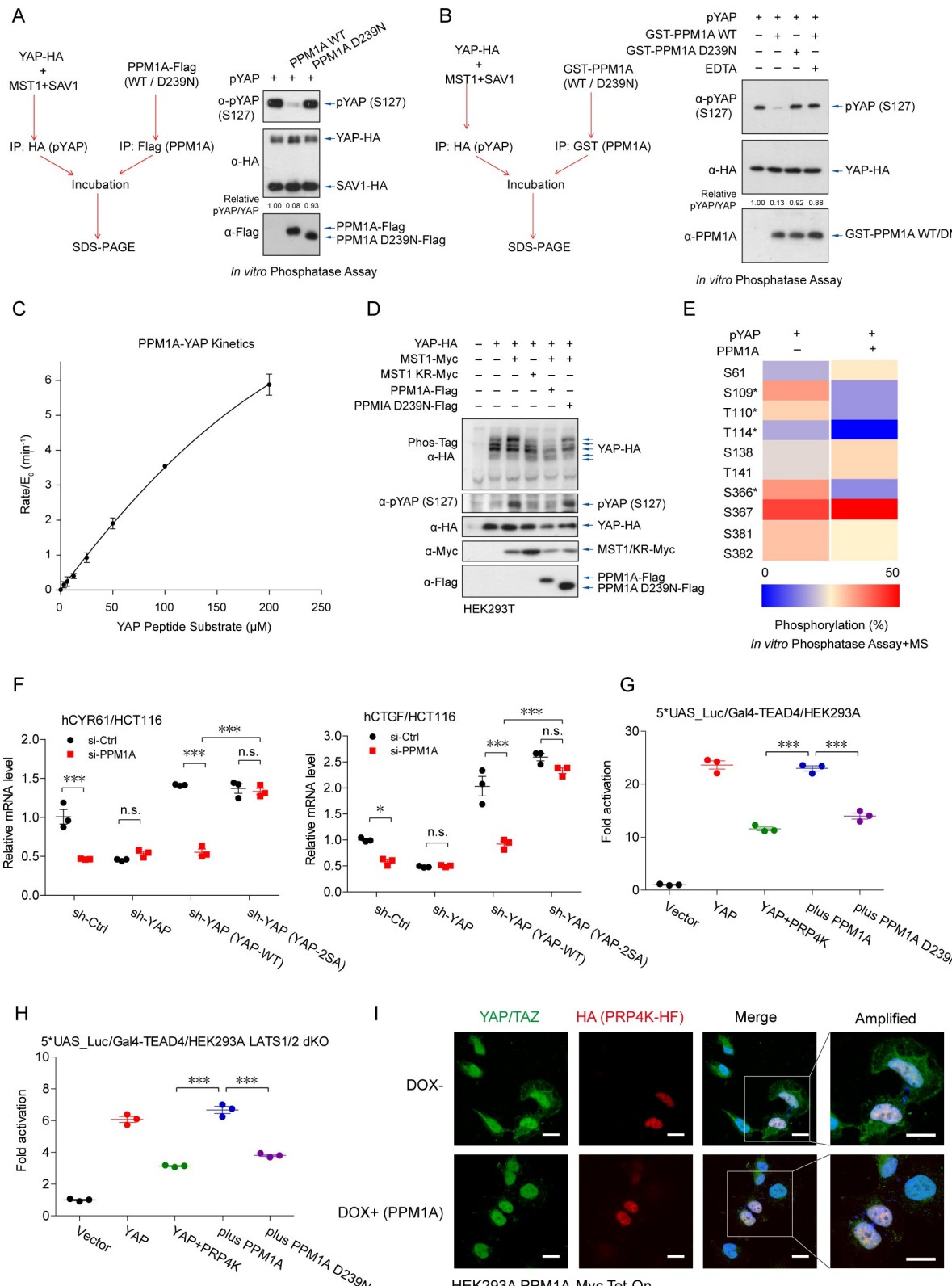

**Fig 4. PPM1A directly and selectively eliminates YAP phosphorylation. (A, B)** YAP was cotransfected together with MST1 and SAV1 and purified by the HA-tag to obtain the phospho-YAP proteins, which was incubated with separately purified PPM1A from cells (A) or

from *E. coli* (B), to perform the in vitro phosphatase assays. Immunoblotting of the assay products exhibited an effective elimination of phospho-YAP (S127) by PPM1A, but not by the phosphatase-dead PPM1A (D293N) or in the presence of EDTA. (**C**) Enzymatic reaction kinetic coefficient was measured of PPM1A against the substrate YAP peptide (sequence of QHVRAH-pSer-SPASLQ) using the malachite green assay. The reaction curve was fitted the Michaelis–Menten equation to calculate a specificity constant. (**D**) PPM1A selectively eliminated a subset of YAP phosphorylation in cells, as shown by the mobility shift using Phos-Tag electrophoresis. (**E**) Mass spectrometry analysis for products of the in vitro phosphatase assay showed that PPM1A selectively dephosphorylated a portion of phospho-residues on YAP, including S109 and S366. Unfortunately, peptides containing the S127 residue were barely covered in this mass spectrometry. (**F**) Reintroduction of YAP 2SA mutant (S109A/S127A) into PPM1A-depleted HCT116 cells restored the mRNA levels of CYR61 and CTGF, which were unaffected now by the PPM1A deficiency. (**G**, **H**) Reporter assays using the TEAD-responsive promoter showed that PRP4K, a nuclear localized protein kinase, inhibited the transcription potency of YAP, either in wild-type (G) or LATS1/2 dKO HEK293 cells (H). Coexpression of wild-type PPM1A restored the PRP4K-suppressed YAP activity. (**I**) Immunofluorescence imaging showed the exclusively localization of PRP4K in the nucleus and an obviously cytoplasmic distribution of YAP/TAZ in cells with PRP4K coexpression, which was prevented by PPM1A induction. Unprocessed images of blots are shown in S1 Raw Images. Statistics source data are provided in S1 Data. dKO, double knockout; LATS1/2, large tumor suppressor kinase 1 and 2; MST1, mammalian sterile 20-like kinase 1; phospho-YAP, phosphorylating forms of YAP; PPM1A, protein phosphatase magnesium-dependent 1A; SAV1, protein salvador homolog 1; TAZ, transcriptional coactivator with PDZ-binding motif; TEAD, transcriptional enhanced associate domain; YAP, Yes-associated protein.

Integrity and function of gut epithelium are important to deal with the constant threats from microbial and environmental factors [45]. The Hippo pathway, interplaying with Wnt signaling, is critical for regulation of intestinal regeneration [46–51]. We then examined the effect of PPM1A in intestinal regeneration, first using a model of in vitro intestinal organoids. Striking phenotypes, including a significantly lower level of domain (crypt) formation and diminished crypt structure, were found in intestinal organoids from PPM1A KO mice (Fig 5C and 5D). Using the Edu integration and staining, we also detected an obviously fewer proliferating cells in organoids from PPM1A KO mice (Fig 5E and 5F). Intriguingly, in wild-type intestinal organoids, most proliferating cells were located at the ends of crypts, where YAP/TAZ were nuclear-localized, in sharp distinction to those PPM1A KO cells with clearly cytoplasmic YAP/TAZ (Fig 5F). In agreement with these observations, the mRNA expressions of YAP/TAZ target genes were significantly diminished in PPM1A KO organoids (Fig 5G). Notably, reconstitution of the YAP 2SA mutant (S109A/S127A), which was supposed to resist PPM1A dephosphorylation, entirely restored the molecular pattern of organoids without PPM1A (Fig 5H). In addition, we found that the pharmacological inhibitions of TGF-β signaling, or neutralizing the type I interferon (IFN-I) signaling, two pathways regulated by PPM1A [40,52], failed to rescue the proliferation defects in PPM1A KO organoids (Fig 5I and 5J), as well as lower expression of YAP/TAZ target gene (Fig 5K). These consistent data suggest that PPM1A is a critical regulator for intestinal regeneration in the murine organoid model, via regulating the activity of YAP/TAZ.

## PPM1A is indispensable for murine intestinal regeneration upon colitis

Recent reports illustrated an essential role of YAP/TAZ in colon regeneration upon dextran sulphate sodium (DSS)-induced colitis [53,54]. We thus evaluated functions of the PPM1A-mediated Hippo signaling regulation in DSS-induced colitis by utilizing the PPM1A KO mice. In the late stage of colitis, PPM1A KO mice displayed an increased score of daily disease activity index (DAI) of colitis (Fig 6A) and more severity in body weight loss (Fig 6B), suggesting the presence of more severe inflammations and/or defects in intestinal regeneration. Notably, the majority of PPM1A KO mice failed to survive and died during the late stage of colitis, in a sharp distinction with wild-type mice that were mostly survived (Fig 6C). We next evaluated the recovery of intestinal histological damage post colitis. The markedly lesser integrity of crypts and villus architectures and more severe depletion of goblet cells were observed in PPM1A KO mice,

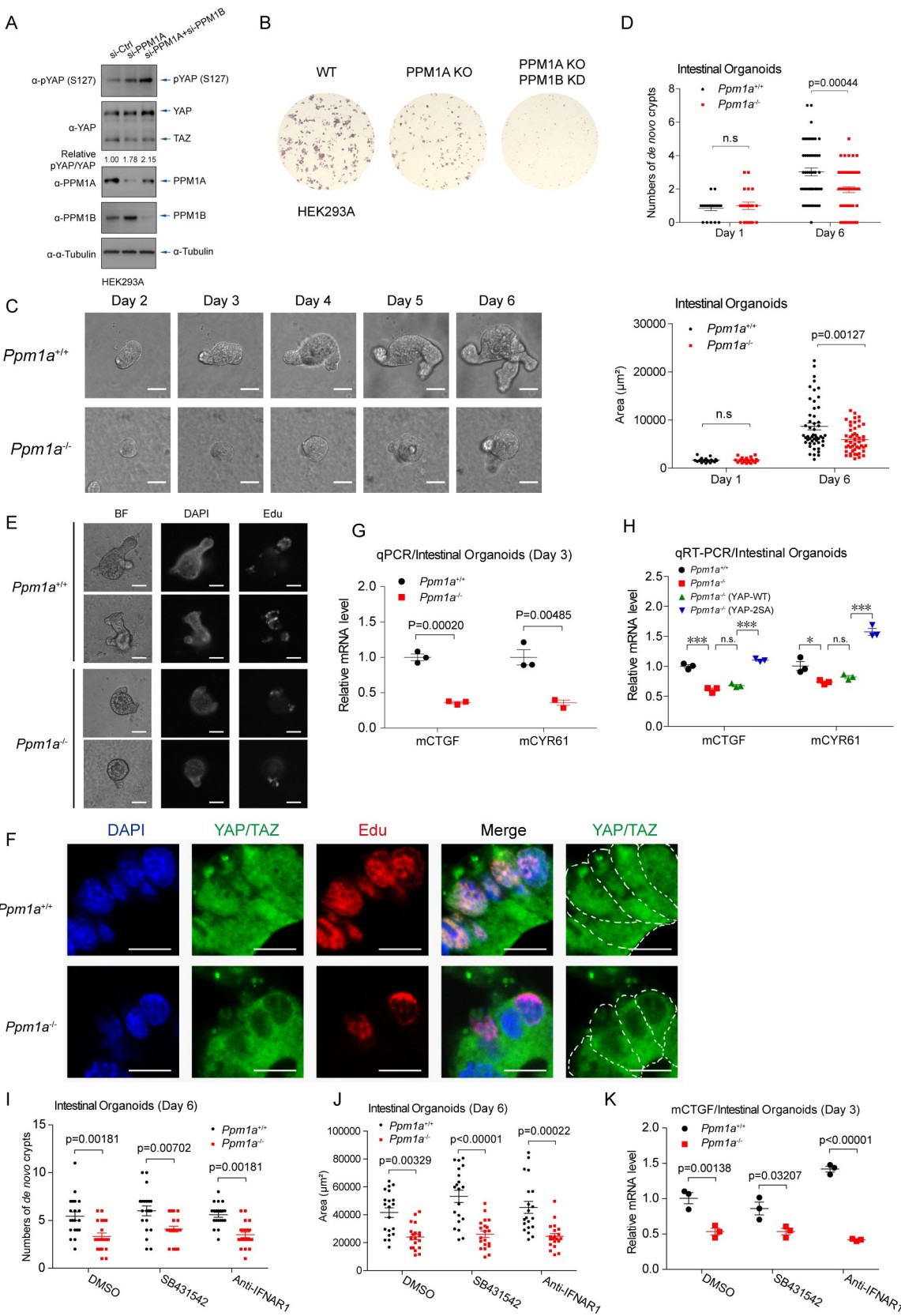

**Fig 5. PPM1A is critical for proliferation of intestinal organoids. (A**, **B)** Double depletion of PPM1A and PPM1B resulted in an enhanced level of phospho-YAP (S127), when compared with depletion of PPM1A alone (A). Depletion of PPM1B by a CRISPR-based strategy in PPM1A KO HEK293 cells led to a complete loss of colony formation (B). **(C, D)** Intestinal crypts were isolated from WT and PPM1A KO mice at 6-week age and cultured in vitro (C). A dramatic lower number of de novo crypts (D, top panel) and the decreased size of organoids (D, bottom panel) were seen in organoids from PPM1A KO mice, with roughly 50 intestine organoids each group examined. **(E)** Edu integration, which mostly localized in the ends of crypts and represented the proliferating cells, was significantly lower in intestinal organoids without PPM1A. **(F)** Anti-YAP/TAZ immunofluorescence imaging of organoid sections detected a substantial subset of cells containing nucleo-YAP/TAZ in the WT crypts and with active proliferation (Edu staining), in a sharp distinction of *Ppm1a*$^{-/-}$ crypts, where YAP/TAZ were located in the cytoplasm even in some proliferating cells. **(G)** The mRNA levels of YAP/TAZ target genes were significantly lower in *Ppm1a*$^{-/-}$ intestinal organoids, examined at Day 3. **(H)** Lentiviral delivery into *Ppm1a*$^{-/-}$ intestinal organoids of the hYAP 2SA mutant (S109A/S127A), which was resistant to the PPM1A-mediated dephosphorylation, restored the mRNA levels of CTGF and CYR61. **(I, J)** The decreased number of de novo crypts (I) and size of organoids (J) in PPM1A KO intestinal organoids were failed to be recovered, when TGF-β signaling was inhibited by small molecule inhibitor SB431542 or when IFN-I signaling was blocked by the neutralizing antibody targeting IFNAR1, examined at Day 6, with roughly 20 intestine organoids each group. **(K)** The mRNA levels of YAP/TAZ target genes in *Ppm1a*$^{-/-}$ intestinal organoids were unable to be restored, under inhibition of TGF-β signaling or IFN-1 signaling, examined at Day 3. Unprocessed images of blots are shown in S1 Raw Images. Statistics source data are provided in S1 Data. IFN-I, type I interferon; KO, knockout; phospho-YAP, phosphorylating forms of YAP; PPM1A, protein phosphatase magnesium-dependent 1A; TAZ, transcriptional coactivator with PDZ-binding motif; TGF-β, transforming growth factor beta; WT, wild-type; YAP, Yes-associated protein.

accompanied with a substantially decrease of proliferating intestinal epithelium (Fig 6D and S4A Fig). PPM1A deficiency resulted in a markedly reduced level of intestinal epithelium by which it contained the nuclear YAP (Fig 6D), and impeded the colon regeneration upon colitis (Fig 6E). Immunohistochemistry (IHC) staining further confirmed the marked decrease of the nucleo-YAP in intestinal cells without PPM1A (Fig 6F). These consistent data, in agreement with the observations from intestinal organoids, suggest that PPM1A, by regulating the YAP cellular distribution and YAP-mediated epithelium proliferation, is critical for intestinal regeneration.

Because PPM1A also functions as a critical negative regulator of TGF-β/Smad signaling and innate nucleic acid sensing [40,52,55,56], we used pharmacological methodology to analyze their individual contributions to these severe colitis phenotypes observed in PPM1A KO mice. The inhibitor of type I TGF-β receptor SB431542, the neutralizing antibody targeting IFN-I receptor, and MST kinase inhibitor XMU-MP-1 [57] were administrated in DSS-induced colitis model. Administration of XMU-MP-1 entirely recovered the severe colitis phenotypes in PPM1A KO mice, including the daily DAI (Fig 6A) and body weight loss (Fig 6B), largely prevented the DSS-induced disruption of crypts and villus architectures in PPM1A KO mice (Fig 6G and S4B Fig), and nicely maintained the proliferating intestinal epithelium (Fig 6G). Inhibition of TGF-β signaling, but not the IFN-I pathway, partially reversed these colitis phenotypes (Fig 6G and S4B Fig), suggesting an involvement but not a key role of TGF-β in intestinal regeneration. Furthermore, blockade of TGF-β signaling or the IFN-I production failed to rescue these severe symptoms of colitis in mice without PPM1A (S4C and S4D Fig). Pharmacological inhibition of Hippo-YAP signaling, but not the inhibition of TGF-β signaling or the IFN-I pathway, entirely survived PPM1A KO mice upon DSS-induced colitis challenge (Fig 6C). To confirm that PPM1A interacts with Hippo signaling genetically, we attempted to examine the colitis phenotypes of *Ppm1a*$^{-/-}$/*Lats1*$^{+/-}$ mice obtained by crossing the PPM1A KO mice with LATS1 heterozygotes. Markedly, attenuated the Hippo signaling by LATS1 deficiency nicely restored the severe colitis phenotypes in PPM1A mice (Fig 6H and S4E Fig). The marked increase of phospho-YAP level was also reverted to normal (Fig 6I), as well as the restored proliferating intestinal epithelium (Fig 6I and S4F and S4G Fig). Taken together, these genetic and pharmacological analyses in murine colitis model suggest an indispensable role of PPM1A in ameliorating the progression of colitis, via PPM1A-mediated suppression of Hippo signaling.

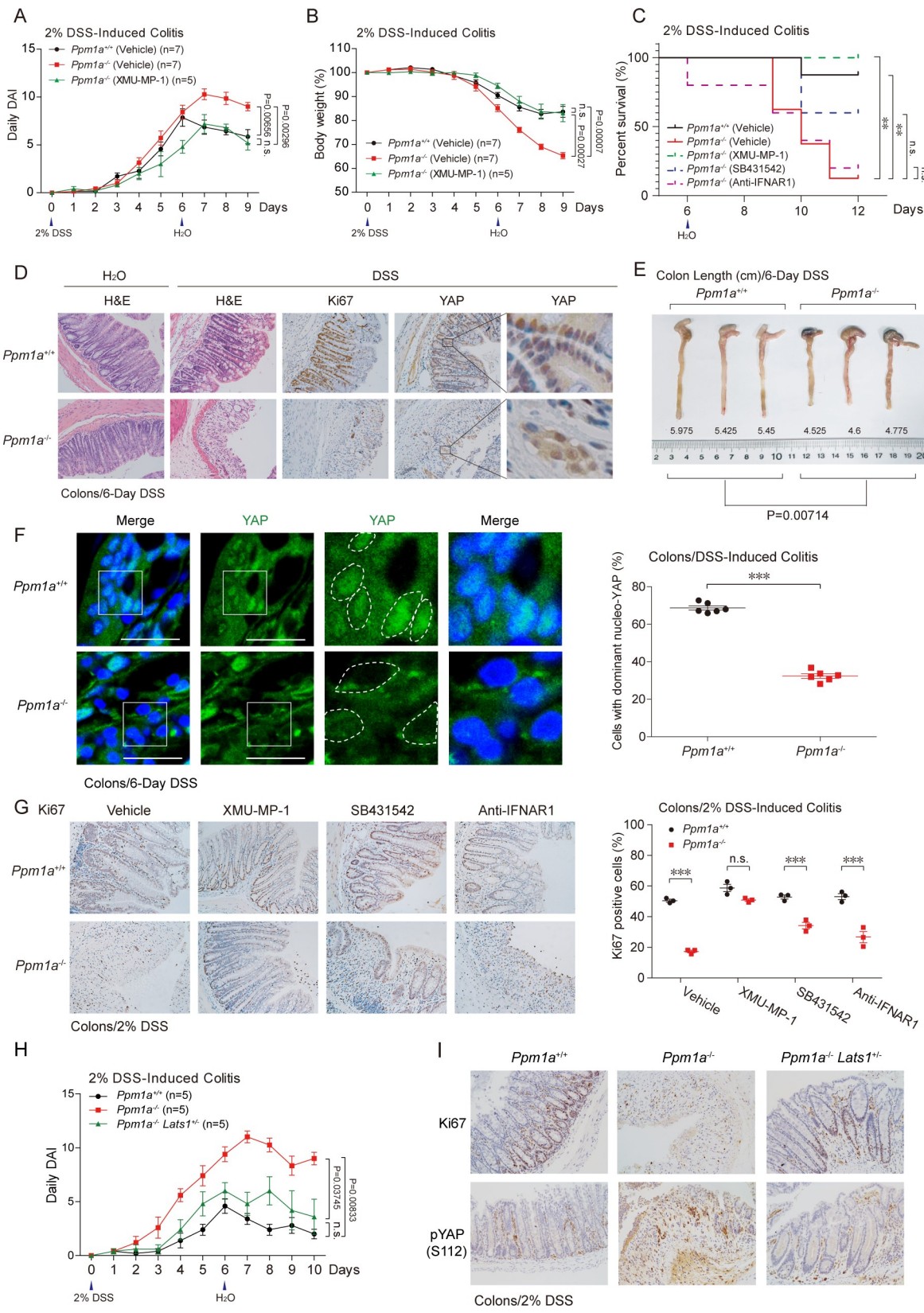

**Fig 6. PPM1A is indispensable for murine intestinal regeneration upon colitis. (A**, **B)** DSS (2%)-induced colitis was performed in wild-type and PPM1A KO mice at 10–12 weeks age. More severe phenotypes of daily DAI (A) and body weight loss (B) were detected in PPM1A KO mice, starting at Day 6. Administration of XMU-MP-1, a small molecular inhibitor of MST kinase, largely restored the colitis phenotypes in $Ppm1a^{-/-}$ mice (A and B). **(C)** DSS-induced colitis resulted in the substantial death of $Ppm1a^{-/-}$ mice, during the extended observation and by log-rank test in statistics. Administration of XMU-MP-1, but not the inhibitors/neutralizing antibody for TGF-β and IFN-I signaling, survived $Ppm1a^{-/-}$ mice. **(D)** DSS treatment led to a severe structural loss of intestine villus, which was largely recovered through intestine regeneration in wild-type mice, as examined at Day 8 (DSS induction for 6 days). The recovery of intestine in $Ppm1a^{-/-}$ mice was markedly compromised, as evidenced by the disorganized intestine villi (second panel, H&E staining) and the obviously decreased number of proliferating cells (Ki67 positive) (third panel, Ki67 IHC). A compromised level of the neucleo-YAP was also seen in $Ppm1a^{-/-}$ intestine (fourth and fifth panels, YAP IHC). **(E)** The length of mice colons post DSS treatment indicated a more severe impairment of intestinal regeneration and/or inflammatory responses in $Ppm1a^{-/-}$ mice at Day 8 (DSS induction for 6 days). **(F)** Immunofluorescence imaging showed a dramatic difference of YAP cellular distribution in intestine villi from wild-type and PPM1A KO mice. A substantial decrease of the nucleo-YAP was seen in intestines from $Ppm1a^{-/-}$ mice. **(G)** Administration with XMU-MP-1 (1 mg/kg, once a day) via IP injection restored both the villi structure (left panel) and percentage of the proliferating cells (right panel) in $Ppm1a^{-/-}$ intestines. Administration of SB431542 (4.2 mg/kg, once a day) alleviated these intestine phenotypes but anti-IFNAR1 (0.2 mg/mouse, once every 2 days) not. **(H, I)** Genetic deficiency of LATS1 ($Lats1^{+/-}$) largely restored the severe colitis phenotypes upon PPM1A deficiency, as indicated by the alleviated level of disease DAI (H) and villi structure that was mostly intact (I). An increase of phospho-YAP (S112) (equivalent human S127) was seen in the absence of PPM1A, which was down-regulated by LATS1 deficiency (I). Unprocessed images of blots are shown in S1 Raw Images. Statistics source data are provided in S1 Data. DAI, disease activity index; DSS, dextran sulphate sodium; H&E, hematoxylin and eosin; IFN-I, type I interferon; IHC, immunohistochemistry; IP, intraperitoneal; KO, knockout; LATS1, large tumor suppressor kinase 1; MST, mammalian sterile 20-like kinase; phospho-YAP, phosphorylating forms of YAP; PPM1A, protein phosphatase magnesium-dependent 1A; TGF-β, transforming growth factor beta; YAP, Yes-associated protein.

## PPM1A deficiency compromises liver regeneration

Upon surgical resection or injury, the Hippo-YAP pathway and fine-tuning of YAP/TAZ are important to control the quiescent state transition and proliferation of hepatocytes [58,59]. We thereby checked hepatocyte proliferation in livers from 10-day postnatal mice with PPM1A deficiency. A compromised level of hepatocyte proliferation was seen in the absence of PPM1A (Fig 7A). We also performed hepatectomy surgery to evaluate the effect of PPM1A in liver regeneration upon injury. Notably, we observed a substantially lower degree for the recovery of liver/body weight (Fig 7B) and the down-regulation of compensatory hepatocyte proliferation (Fig 7C), in PPM1A KO mice at 3 to 4 day post hepatectomy (dph). In addition, nucleo-YAP/TAZ was abundantly detected in sections on compensatory growth livers from wild-type mice but was very scarce in similar liver sections from PPM1A KO mice (Fig 7D). Similar to the phenotypes in intestinal epithelium, blockade of TGF-β signaling or IFN-I productions failed to reverse the compromised phenotypes of hepatocyte proliferation in PPM1A KO mice (Fig 7E). In contrast, administration of MST inhibitor, which down-regulated the level of phospho-YAP (Fig 7F), largely restored the hepatocyte proliferation in $Ppm1a^{-/-}$ livers (Fig 7G). These consistent observations in $Ppm1a^{-/-}$ livers suggest that PPM1A is a physiological regulator of Hippo-YAP signaling to facilitate liver regeneration.

## Discussion

The Hippo pathway drives TEAD-mediated transcriptome as well as noncanonical mechanisms, such as YAP/TAZ-mediated regulation of innate and adaptive immunity [24,60,61]. A phosphorylation cascade within a core kinase-chain constitutes the molecular basis of this highly conserved signaling mechanism, which regulates the nucleocytoplasmic translocation and stability of YAP/TAZ. However, physiological functions of dephosphorylation event in mammalian Hippo-YAP signaling are still barely known. Particularly, there is lacking of genetic and in vivo evidence as to the existence of bona fide phosphatase(s) that directly eliminate the key phosphorylation modifications of YAP/TAZ and determine their fate. In this report, we identify PPM1A, a metal-dependent protein phosphatase known for regulating immune responses [40,52,56,62,63] and stress signaling [64,65], is a phosphatase directly effects on YAP/TAZ.

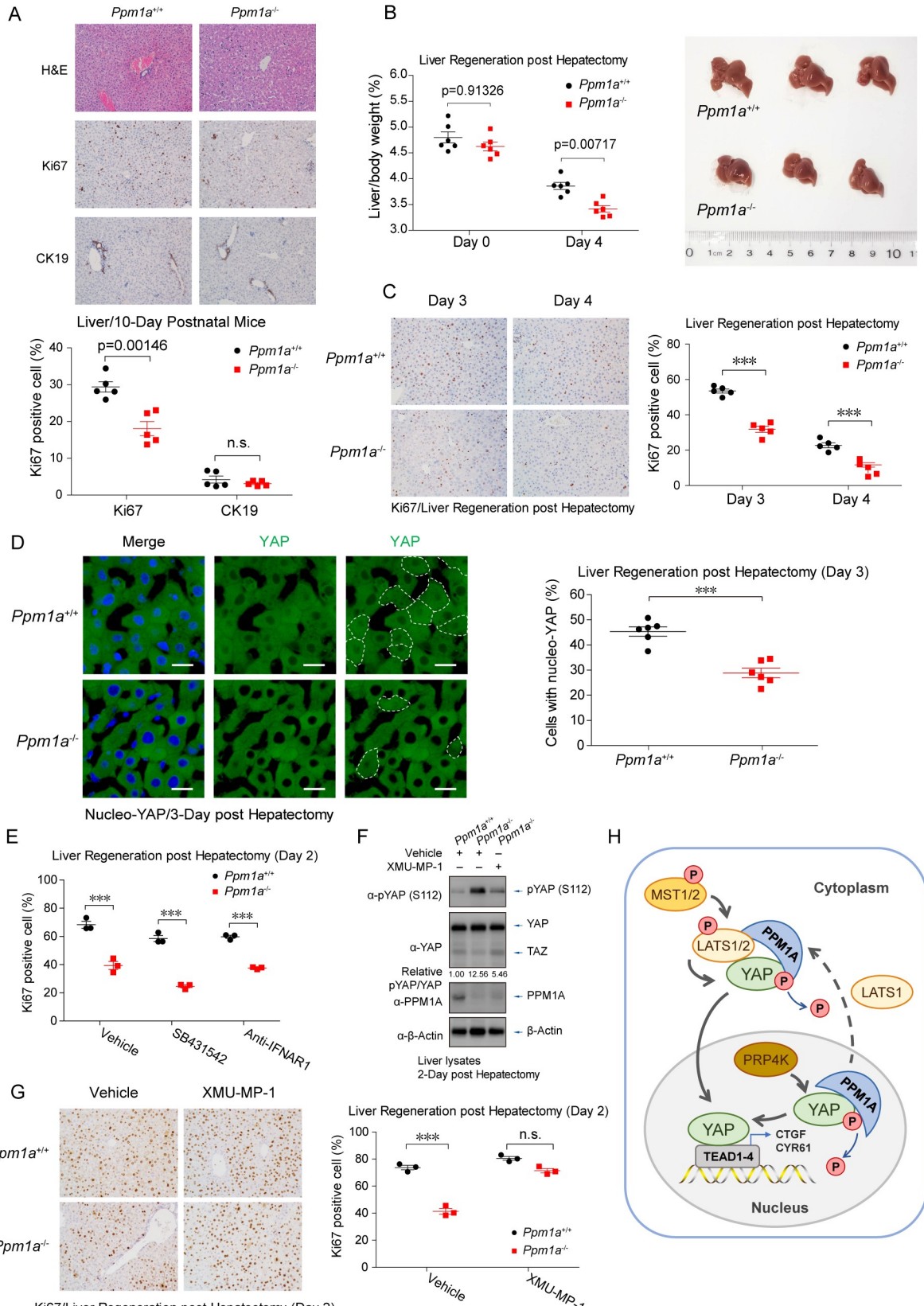

**Fig 7. PPM1A deficiency impedes the liver regeneration. (A)** H&E staining of liver sections from the 10-day postnatal mice showed a reduced level of hepatocyte proliferation (Ki67 positive) but an elevated level of immune cell infiltration and vacuoles in PPM1A KO mice. **(B)** Wild-type and PPM1A KO mice at 8–10 weeks age were carried out with hepatectomy surgery. The ratio of liver/body weight was measured at day 4 post hepatectomy, which exhibited a marked decrease of compensatory liver regeneration in *Ppm1a⁻/⁻* mice. *n* = 6 mice in each group. **(C)** A compromised level of hepatocyte proliferation in compensatory liver regeneration of *Ppm1a⁻/⁻* mice was detected by Ki67 staining examined at day 3 and day 4 post hepatectomy. **(D)** Immunofluorescence imaging of liver sections revealed a dramatic decrease of the nucleo-YAP in livers from *Ppm1a⁻/⁻* mice examined at day 3 post hepatectomy, exhibiting by the fewer YAP proteins in the nucleus (darker imaging in the nucleus, left panel) and the substantially lesser cells with the nucleo-YAP localization (circled, and right panel). **(E)** Administration of SB431542 or anti-IFNAR1 failed to restore the suppressed hepatocyte proliferation in compensatory liver regeneration in *Ppm1a⁻/⁻* mice, examined at Day 2. **(F, G)** Administration of MST inhibitor XMU-MP-1 decreased the level of phospho-YAP (S112) (F) and increased the proliferating hepatocyte (G) in *Ppm1a⁻/⁻* livers. **(H)** A schematic model for the PPM1A-regulated Hippo-YAP signaling. PPM1A associates with YAP/TAZ both in the cytoplasm and in the nucleus and directly eliminates the key phosphorylation modifications on YAP, which determines the nucleocytoplasmic localization and transcription potency of YAP/TAZ. Accordingly, PPM1A functions as a critically physiological regulator of the Hippo-YAP pathway in mammals, including the intestinal and liver regeneration upon injuries. Unprocessed images of blots are shown in S1 Raw Images. Statistics source data are provided in S1 Data. H&E, hematoxylin and eosin; KO, knockout; MST, mammalian sterile 20-like kinase; phospho-YAP, phosphorylating forms of YAP; PPM1A, protein phosphatase magnesium-dependent 1A; TAZ, transcriptional coactivator with PDZ-binding motif; YAP, Yes-associated protein.

Previous studies have reported that PPM1A terminates TGF-β signaling by dephosphory-lating and inactivating Smad2/3, the essential effectors of the TGF-β pathway [52,55]. PPM1A also functions as a key negative regulator of innate nucleic acid sensing to effect on the mito-chondrial antiviral signaling protein (MAVS) and stimulator of interferon genes (STING) sig-nalosome [40,56,66], which suppresses the production of type I interferons (IFNs) and attenuates the antiviral immunity. *Ppm1a⁻/⁻* mice exhibit the lasting inflammatory responses in skin wound healing [67]. Here, we found that PPM1A dominated the nucleocytoplasmic translocation and physiological function of YAP/TAZ in mammalian intestinal and liver regeneration, by directly eliminating YAP phosphorylation at the critical S127 residue (Fig 7H). The PPM1A-dominated distribution of YAP/TAZ and subsequent target genes transcrip-tion are very clear, supported by consistent observations at cellular, tissue, and whole-animal levels. Dephosphorylating event of YAP also played a critical role in cellular responses to nutri-ent and energy status. Recently, the pivotal roles of Hippo-YAP signaling are recognized in regeneration of gut and liver [46,48,49,51,58,59]. Intriguingly, we revealed now that a necessity of PPM1A and YAP dephosphorylation event in these critical biological processes, as evi-denced by various loss- and gain-of-function investigations in organoids and murine models.

## Mechanism for PPM1A-mediated quiescence of Hippo signaling

Our current observations showed that PPM1A endogenously interacted and targeted on YAP to selectively eliminate phosphorylation on several residues, including the phospho-S127 that dictates its cellular localization. We also demonstrated PPM1A as a Ser/Thr phosphatase that directly and effectively dephosphorylate YAP, based on in vitro enzymatic analyses with puri-fied substrate and recombinant PPM1A from the cells and bacteria. However, it is probable that PPM1A also targets other key components of the Hippo signaling complexes, including the MST-SAV, LATS-MOB, and YAP-TEAD complexes, but requiring further validation.

Because PPM1A functions an efficient phosphatase to eliminate the key phosphorylation of YAP/TAZ, we have observed a profound alteration of YAP/TAZ cellular distribution under various gain- or loss-of-function studies. PPM1A preferably distributes in the nucleus in many types of cells [40]. Intriguingly, we also observed an obvious change of PPM1A distribution during the activation of Hippo signaling, which was appeared to be mediated by LATS1 kinase. Accordingly, a PPM1A-LATS1 complex was easily detected. Meanwhile, PPM1A appears to work independently in the nucleus, which eliminating the PRPK4-mediated YAP phosphory-lation [43] and keeping YAP alive. We speculate that the PPM1A-YAP/TAZ regulation in the nucleus is critical. How YAP/TAZ are precisely regulated by intracellular conditions is still not

fully understood, except under a few particular conditions such as GPCR activation, energy stress, and serum starvation [7,13,14,36,68]. Because PPM1A is mobilized by Hippo signaling via LATS1, we speculate that PPM1A serves as a key negative feedback mechanism of the Hippo pathway. An intriguing question is raised as to whether PPM1A can function as a sensor of phospho-YAP. The PPM1A dependence of YAP/TAZ activation indicates an extra level for YAP/TAZ activation and suggests that other signaling pathways which interact with PPM1A, such as the TGF-β pathway and innate sensing pathway, may cooperate with the Hippo-YAP pathway in many biological processes via the PPM1A.

## Functions of PPM1A in Hippo signaling, organ regeneration, and colitis

We propose that the reverse phosphorylation event of Hippo-YAP machinery by PPM1A serves an adaptive mechanism to ensure an elaborate equilibrium in response to dynamic cellular environments. PPM1A is considerably integrated in a few pivot signaling pathways including TGF-β [52,55], innate nucleic acid sensing [40, 56], NF-κB [62,63], and p38 and JNK MAPK [64]. As a result, mutual interactions of these key mechanisms by PPM1A, including the Hippo pathway, may together contribute to efficient and combinational effects for adaptions of specific physiological or pathological contexts. Notably, we have observed the dramatic inflammatory phenotypes of gut in PPM1A-ablated mice, in well agreement with observations from our and other groups that YAP/TAZ are important suppressors to innate immunity for NF-κB activation [25] and IFN-I production [24], as well as the enhancers for suppressive Treg lymphocytes [61]. This opinion is also supported by severe inflammatory phenotypes and elevated antitumor immunity in mice with YAP and/ or TAZ deficiency in various settings [61,69–72]. Therefore, loss-of-function of PPM1A relieves the YAP/TAZ-mediated suppression of innate and adaptive immunity, which leads to robust infiltration of immune cells into the organ, similar to a recently reported regulation [73]. It is even surprising that PPM1A is so critical for the regeneration of intestines and livers and disease phenotypes of colitis. Beside strong phenotypes of regeneration defects in intestine and liver, we also observed the clearly lesser distribution of YAP/TAZ in the nucleus, in intestine, and in liver without PPM1A, which also supports a direct and indispensable involvement of PPM1A in the function of Hippo signaling. Furthermore, a specific inhibitor of MST kinases [57], or half-deficiency of LATS1, well rescues the defective phenotype in PPM1A KO mice, demonstrating that the key role of the PPM1A-YAP axis in guts and livers and diseases such as colitis.

In conclusion, our study provides in vivo and genetic evidence to support a critical role of PPM1A and the reverse phosphorylation event of YAP/TAZ in regulation of the Hippo pathway in mammalian. Our model suggests that the level and activity of PPM1A functions as a critical physiological determinant of Hippo-mediated extra- to intracellular communications. Consistent with this notion, our study proposes that pharmacological suppression of PPM1A, such as by membrane-permeable molecules, may offer a potential therapeutic benefit for YAP/TAZ-related diseases.

## Methods

### Expression plasmids, reagents, and antibodies

Expression plasmids encoding Flag-, Myc-, or HA-tagged wild type or mutants, of human PPM1A, YAP, MST1, LATS1, LATS2, TAZ, MOB1, SAV1, and reporters of 5×UAS_Luc and Gal4-TEAD4_Luc have been described previously [24,40,41,74]. Expression plasmids for MAP4K1 and PRP4K were obtained from the kinase library from Life Sciences Institute (Zhejiang University, Hangzhou). GST-PPM1A, GST-PPM1A D239N were constructed on pGEX-

4T-1 vector. The cDNA library of Ser/Thr phosphatases was constructed and described previously [40]. All coding sequences were verified by DNA sequencing. The list of recombinant DNA is provided in the attached S1 Table.

The pharmacological reagents 2-DG (Sangon Biotech, Shanghai, China), Dox (Sangon Biotech), Edu (RiboBio, Guangzhou, China), puromycin (Yeasen, Shanghai, China), and G418 (Yeasen) were purchased. Detailed information of all the antibodies applied in immunoblotting, immunoprecipitation, immunofluorescence, and immunohistochemistry are provided in the attached S2 Table. The antibodies included anti-YAP/TAZ (sc-101199, Santa Cruz Biotechnology, Dallas, Texas, United States of America, 1:1,000 dilution), anti-YAP (14074, Cell Signaling Technology, Danvers, Massachusetts, USA, 1:1,000 dilution), anti-pYAP (S127) (4911, Cell Signaling Technology, 1:1,000 dilution), anti-pYAP (S127) (13008, 1:1,000 dilution, Cell Signaling Technology), anti-LATS1 (3477, Cell Signaling Technology, 1:1,000 dilution), anti-pLATS1 (T1079) (8654, Cell Signaling Technology, 1:1,000 dilution), anti-pTAZ (S89) (59971S, Cell Signaling Technology, 1:1,000 dilution), anti-pTBK1 (S172) (5483S, Cell Signaling Technology, 1:2,000 dilution), anti-PPM1A (3549, Cell Signaling Technology, 1:2,000 dilution), anti-Myc (2276S, Cell Signaling Technology, 1:3,000 dilution) and anti-HA (3724S, Cell Signaling Technology, 1:3,000 dilution), anti-α-tubulin (T6199-200UL, Sigma-Aldrich, St. Louis, Missouri, USA, 1:20,000 dilution), anti-β-Actin (A5441, Sigma-Aldrich, 1:20,000 dilution) and anti-Flag (M2) (F3165-5MG, Sigma-Aldrich, 1:3,000 dilution), anti-Ki67 (GB13030-2, Servicebio, Wuhan, China, 1:200 dilution), and anti-CK19 (GB14058, Servicebio, 1:100 dilution).

## Cell culture, transfections, and infections

HEK293T, HEK293A, HaCaT, A549 cells were from ATCC. MST1/2 dKO and LATS1/2 dKO HEK293A were gifted from Dr. Kun-Liang Guan (University of California at San Diego). Primary MEFs were obtained from E12.5 to E13.5 embryos in pregnant C57BL/6 female mice at 8 weeks of age. Primary TTFs were obtained from C57BL/6 male mice at 6 to 8 weeks of age. No cell lines used in this study were found in the database of commonly misidentified cell lines that is maintained by ICLAC and NCBI Biosample. Cell lines were frequently checked for morphology under a microscope and tested for mycoplasma contamination but were not authenticated. HEK293T, HEK293A, A549 cells, MEFs, TTFs, and HCT116 were cultured in DMEM medium with 10% fetal bovine serum (FBS) at 37˚C in 5% $CO_2$ (v/v), and HaCaT cells were cultured in MEM medium with 10% FBS. The PPM1A inducible expressing HEK293A cells were generated by lentiviral vector containing the inducible Tet-On system followed by ORF of PPM1A, and selected by G418 antibiotic at concentration of 1,500 μg/ml for 7 days. LipofectAmine 3000 (Invitrogen, Thermo Fisher Scientific, Waltham, Massachusetts, USA) or Polyethylenimine (PEI, Polysciences, Warrington, Pennsylvania, USA) transfection reagents were used for plasmid transfection.

## Luciferase reporter assay

HEK293 cells were transfected with the indicated reporters (100 ng) bearing an ORF coding Firefly luciferase, along with the pRL-Luc with Renilla luciferase ORF as the internal control for transfection and other expression vectors specified in the results section. Cells were treated with the indicated conditions and lysed at 24 hours after transfection in a passive lysis buffer (Promega, Madison, Wisconsin, USA). Luciferase assays were performed using a dual luciferase assay kit (Promega), quantified with POLARstar Omega (BMG Labtech, Ortenberg, Germany), and normalized to the internal Renilla luciferase activity.

## CRISPR/Cas9-mediated generation of PPM1A KO cells

CRISPR/Cas9 genomic editing for gene deletion was performed as described [75]. Guide RNA (gRNA) sequences targeting PPM1A and PPM1B exon were cloned into the pX330 plasmids. These constructs together with the puromycin vector pRK7-puromycin were transfected into HEK293A at a ratio of 15:1 using LipofectAmine 3000 transfection reagent. Twenty-four hours after transfection, cells were selected by puromycin (1.5 μg/ml) for 72 hours, and single clones were obtained by serial dilution and amplification. Clones were identified by immunoblotting with anti-PPM1A antibody, and 2 individual clones of PPM1A$^{-/-}$ were randomly selected and used for the indicated analyses. All gRNAs used in the experiments are also listed in S3 Table.

## siRNA or shRNA-mediated RNA interference

Double-stranded siRNA (RiboBio) to silence endogenous PPM1A in HEK293T, HaCaT, A549 cells targeted the human PPM1A mRNA (sequence information is in S3 Table). Control siRNA (RiboBio) was used to control for possible nonspecific effects of RNA interference. Cells were transfected with siRNA using the Lipofectamine RNAiMAX (Invitrogen) reagent for 48 hours before the further assay, and the reverse transfection method was used to reach optimal efficiency. The shRNA-mediated knockdown of YAP in HCT116 cells was generated by shRNAs (sequence information is in S3 Table), delivered by the lentiviral vector produced by the Mission shRNA (Sigma-Aldrich) plasmids (TRCN information is in S3 Table), together with pMD2.G and psPAX2 plasmids in 293T cells.

## Colony formation assay

HEK293A cells were planted in each well of the 6-well plate with a concentration of $3 \times 10^3$ cells/well and cultured in a 5% $CO_2$ humidified incubator at 37˚C for 1 to 2 weeks. The colonies were stained with crystal violet for observation.

## Culture and characterization of intestinal organoids

Intestinal organoids were cultured according to a previously described protocol established by Sato and Clevers [76]. Isolated small intestines were opened longitudinally, washed with cold PBS for 3 times, and villi were scraped. Tissue fragments were then incubated in 2 mM EDTA with PBS for 15 minutes on ice for twice, added into 50 ml cold PBS and shake fiercely, and passed through a 70-μm cell strainer (BD Bioscience, San Jose, California, USA) to remove residual villous material. Isolated crypts were centrifuged at 300$g$ for 5 minutes to separate crypts from single cells, which was suspended in matrigel for in vitro culture, overlaid with organoid culture ENR medium [45]. Upon plating, organoid formation was analyzed each day, and medium was changed every 2 days. De novo crypts/domains were scored as any protrusions, typically containing paneth cells, budding from the initial sphere formed after seeding crypts. Crypts were counted from bright-field images using Image J software (National Institute of Mental Health, Bethesda, Maryland, USA).

## Infection of intestinal organoids by lentivirus

HEK293T were cultivated and transfected the lentivirus plasmids pLVX, pLVX-YAP-WT, or pLVX-YAP-S109/127A. The supernatant was collected after 24 to 48 hours of infection and concentrated by ultracentrifugation. Organoids cultured in ENRwntNic medium (ENR medium with Wnt3a and Nicotinamide) for at least 3 days. Organoids were infected 6 hours with the concentrated lentiviruses produced from HEK293T cells and reseeded in the fresh

ENR medium. HCT116 cells were infected by the concentrated lentiviruses for 2 days and harvested for the quantitative real-time PCR (qRT-PCR) assays.

## Quantitative RT-PCR assay

The HEK293, HaCaT, TTFs, HCT116, or organoids were lysed, and total RNA was extracted using RNAeasy extraction kit (Axygen, Union City, California, USA). cDNA was generated by one-step iScript cDNA synthesis kit (Vazyme, Nanjing, China), and qRT-PCR was performed using EvaGreen Qpcr MasterMix (Abm, Vancouver, Canada) and a CFX96 real-time PCR system (Bio-Rad, Hercules, California, USA). Relative quantification was expressed as $2^{-\Delta Ct}$, where $C_t$ is the difference between the main $C_t$ value of triplicates of the sample and the $C_t$ value of endogenous L19 mRNA. The human or mouse primer sequences used are listed as follows:

hCTGF-Forward Primer 5′—CCAATGACAACGCCTCCTG—3′
hCTGF-Reverse Primer 5′—TGGTGCAGCCAGAAAGCTC—3′
hCYR61-Forward Primer 5′—AGCCTCGCATCCTATACAACC—3′
hCYR61-Reverse Primer 5′—TTCTTTCACAAGGCGGCACTC—3′
hYAP-Forward Primer 5′—TAGCCCTGCGTAGCCAGTTA—3′
hYAP-Reverse Primer 5′—TCATGCTTAGTCCACTGTCTGT—3′
mCTGF-Forward Primer 5′—CAAGGACCGCACAGCAGTT—3′
mCTGF-Reverse Primer 5′—AGAACAGGCGCTCCACTCTG—3′
mCYR61-Forward Primer 5′—GCTCAGTCAGAAGGCAGACC—3′
mCYR61-Reverse Primer 5′—GTTCTTGGGGACACAGAGGA—3′.
All primers used in qRT-PCR assay were also attached in S3 Table.

## Coimmunoprecipitations and immunoblottings

Liver lysates upon hepatectomy surgery and HEK293 cells with indicated treatments were lysed using a modified Myc lysis buffer (MLB) (20 mM Tris-Cl, 200 mM NaCl, 10 mM NaF, 1 mM NaV$_2$O$_4$, 1% NP-40, 20 mM β-glycerophosphate, and protease inhibitor (pH 7.5)) [77]. Cell lysates were then subjected to immunoprecipitation using antibodies of anti-Flag (Sigma-Aldrich, F3165-5MG, 1:200 dilution), anti-Myc (2276S, Cell Signaling Technology, 1:200 dilution), or anti-HA (3724S, Cell Signaling Technology, 1:200 dilution) for transfected proteins, or using anti-PPM1A (3549, Cell Signaling Technology, 1:100 dilution) antibody for endogenous proteins. After 3 to 4 washes with MLB, adsorbed proteins were resolved by SDS-PAGE (Bio-Rad) and immunoblotting with the indicated antibodies. Cell lysates were also analyzed using SDS-PAGE and immunoblotting to control the protein abundance.

## Purification of GST-tagged recombinant proteins

PPM1A and its D239N mutant were cloned into the pGEX-4T-1 vector and transformed in DE3 (BL21). The expression of GST-PPM1A WT/DN was induced by IPTG (0.2 mM) at 4°C for 12 hours, and DE3 were suspended in 0.5 ml of GST lysis buffer (50 mM Tris-HCl, 300 mM NaCl, 10 mM MgCl$_2$, 1 mM DTT, 1 mM Pic, 1 mM PMSF (pH 7.5)), sonicated and centrifuged for 12,000$g$ at 4°C for 20 minutes. The supernatants were examined for expression of indicated proteins by SDS-PAGE and Coomassie blue staining.

## In vitro phosphatase assay

The procedure of in vitro phosphatase assay was as previously described [40]. HEK293T cells were transfected with plasmids for expression of indicated proteins, including Myc-MST1,

HA-SAV1, HA-YAP, HA-TAZ, Flag-TBK1, Flag-PPM1A, or Flag-PPM1A D239N. Cells were lysed in modified MLB lysis buffer after 36 hours of transfection, and immunoprecipitations were performed by using anti-Flag (F3165-5MG, Sigma-Aldrich, 1:200 dilution), or anti-HA (3724S, Cell Signaling Technology, 1:200 dilution) antibodies. With twice washes in MLB and 2 washes in phosphatase assay buffer (20 mM Tris-HCl, 1 mM EGTA, 5 mM MgCl₂, 0.02% 2-mercaptoethanol, 0.03% Brij-35, BSA (0.2 mg/ml)), immunoprecipitated HA-YAP and Flag-PPM1A, or recombinant PPM1A from the *E. coli*, were incubated in phosphatase assay buffer at 25°C for 60 minutes on a THERMO-SHAKER for the phosphatase assay. EDTA (1 mM), which deprived the $Mg^{2+}$/$Mn^{2+}$ in the reaction system, was added as a negative control. The reaction was quenched by the addition of 2×SDS loading buffer, and the samples were subjected to SDS-PAGE and specified immunoblotting.

## Malachite green phosphatase assay

Michaelis–Menten steady-state kinetic analysis of PPM1A (a.a. 1–297) was performed by incubating the recombinant phosphatase domain of PPM1A with different concentrations of the phosphopeptide QHVRAH-pSer-SPASLQ. The reaction was carried out by incubating 230 nM PPM1A with peptide concentrations ranging from 0 μM to 200 μM in a phosphatase reaction buffer (50 mM Tris-CL (pH 7.5), 40 mM NaCl, 10 mM β-mercaptoethanol (BME) and 30 mM MgCl₂) at 30°C for 20 minutes. The reaction was quenched by adding 20 μl of the reaction mixture to 40 μl of Biomol Green Reagent (Part of the Biomol Green phosphate detection assay from Enzo Life Sciences, Farmingdale, New York, USA), which was subsequently incubated for 15 minutes to allow for color development. Absorbance measurements were carried out in the Tecan Infinite M200 microplate reader at the wavelength of 620 nm in a 96-well flat transparent microplate. A standard curve for signal versus phosphate concentration was determined by using phosphate standard for Biomol green at a concentration ranging from 0 μM to 10 μM. The phosphate released during the enzymatic catalysis of PPM1A in the reaction was determined from the standard curve, and reaction rates were plotted against the substrate concentration and fit to the equation $\frac{\text{Rate}}{\text{Eo}} = \frac{k_{on}[S]}{1 + \frac{k_{on}}{k_{cat}}[S]}$ on the KaleidaGraph software (Synergy Software, Reading, Pennsylvania, USA) to get the value of the specificity constant $k_{on} = k_{cat}/K_m$ of the reaction.

## Nano-liquid chromatography–tandem mass spectrometry analysis

Nanoscale liquid chromatography coupled to tandem mass spectrometry analysis for protein identification, characterization, and label-free quantification was performed by Phoenix National Proteomics Core services. The samples for mass spectrometry analyses were from the in vitro phosphatase assays mentioned above, separated by SDS-PAGE, stained by Coomassie blue to locate the YAP, and cut spanning YAP. Pryptic peptides were analyzed by Q Exactive HF-X (Thermo Scientific instrument model). Proteins were identified using the search engine of the National Center for Biotechnology Information against the human or mouse RefSeq protein databases. The results of mass spectrometry analysis are listed in S4 Table. The MS raw files have been deposited to the Mass Spectrometry Interactive Virtual Environment (MassIVE MSV000086694).

FTP Download Link: ftp://massive.ucsd.edu/MSV000086694/

## Immunofluorescence and microscopy

To visualize the subcellular localization of inducible or endogenous PPM1A and YAP, or transfected YAP-HA, LATS1-HA, and PRP4K-HF, HEK293A cells were treated as indicated

or transfected with specified plasmids for 24 hours before harvest. In the case of Edu staining of intestinal organoids, 50 μM Edu (RiboBio) was added to the culture media for 2 hours before fixing organoids. The cells/organoids were fixed in 4% paraformaldehyde, blocked in 2% bovine serum albumin in PBS for 1 hour, and incubated sequentially with primary antibodies anti-HA (3724S, Cell Signaling Technology, 1:200 dilution), anti-Flag (F3165-5MG, Sigma-Aldrich, 1:500 dilution), anti-Myc (2276, Cell Signaling Technology, 1:100 dilution), anti-YAP (sc-101199, Santa Cruz Biotechnology, 1:100 dilution) or anti-YAP (14074, Cell Signaling Technology, 1:100 dilution), or anti-PPM1A (3549, Cell Signaling Technology, 1:200 dilution) and Alexa-labeled secondary antibodies (Jackson ImmunoResearch, West Grove, Pennsylvania, USA, 111-095-003; 115-095-003; 111-025-003; 115-025-003, 1:500 dilution) with extensive washing. Slides were then mounted with VectaShield and stained with DAPI (Vector Laboratories, Burlingame, California, USA). Immunofluorescence images were obtained and analyzed using the Nikon Eclipse Ti inverted microscope or by the Zeiss LSM710 confocal microscope.

### Ethics statement

C57BL/6 mice were maintained under specific-pathogen-free (SPF) conditions, and care of experimental animals was in accordance with guidelines and approved by Laboratory Animal Committee of Zhejiang University (approval number ZJU20170658).

### DSS-induced colitis

*Ppm1a*$^{-/-}$ C57BL/6 mice were generated as previously described [40]. *Lats1*$^{+/-}$ C57BL/6 mice were purchased from Cyagen Biosciences (Suzhou, China). For induction of colitis, 10 to 12 weeks aged male C57BL/6 mice were fed formulated drinking water containing 2% to 3% DSS (m.w. 36 to 50 kDa, Cat. #160110, MP Biochemical, Irvine, California, USA) for 6 days before switching to regular drinking water. The mice were monitored for daily body weights and DAI score, which was based on weight loss, rectal bleeding, and assessment of stool consistency [78]. Indicated mice were intraperitoneal (IP) injected with vehicle, XMU-MP-1 (1 mg/kg, once a day, S8334, Selleck, Houston, Texas, USA), SB431542 (4.2 mg/kg, once a day, S1067, Selleck), and Anti-IFNAR1 (0.2 mg/mouse, once every 2 days, BE0241, Bio X Cell, Lebanon, New Hampshire, USA) throughout the study. The mice were euthanized and colons were harvested on the eighth or tenth day.

### Partial hepatectomy

Eight to ten weeks aged *Ppm1a*$^{-/-}$ and littermate *Ppm1a*$^{+/+}$ C57BL/6 mice were anesthetized via an IP injection of pentobarbital sodium in PBS (55 mg/kg body weight) and underwent a two-thirds partial hepatectomy through the resection of the median and left lateral hepatic lobes [79]. The regenerating livers were harvested at the indicated time post-surgery to measure the liver/body weight ratio (%) and the percentage of Ki67 positive cells. The mice were injected with the indicated drugs or the vehicle control via an IP injection at 2 days before surgery. The regenerating livers were pulverized by a ball mill and lysed by a modified MLB [77] for SDS-PAGE and immunoblotting.

### Histology and immunohistochemistry

For DAB immunohistochemistry assays, liver or colon samples from mice were dissected and fixed in 4% paraformaldehyde for 24 hours at 4˚C, dehydrated by EtOH with series of concentration, embedded in paraffin wax block, and the 6 μm sections were then stained by hematoxylin and eosin (H&E), and incubated sequentially with primary antibodies including anti-Ki67

(GB13030-2, Servicebio, 1:200 dilution), anti-CK19 (GB14058, Servicebio, 1:100 dilution), anti-YAP (14074, Cell Signaling Technology, 1:100 dilution), anti-pYAP (S127) (13008, Cell Signaling Technology, 1:200 dilution). For fluorescence immunohistochemistry assays, colon samples were dissected and fixed in 4% paraformaldehyde for 12 hours at 4˚C, dehydrated with 30% sucrose overnight at 4˚C, embedded in optimal cutting temperature compound, and immediately frozen at −80˚C. Sectioned samples at 10 μm were washed 2 times with PBS and permeabilized with 0.5% Triton X-100 for 5 minutes, blocked in 3% bovine serum albumin in PBS for 30 minutes, and incubated sequentially with primary antibodies including anti-YAP (14074, Cell Signaling Technology, 1:100 dilution). After incubation with Alexa-labeled secondary antibodies (Jackson ImmunoResearch, 111-095-003, 1:500 dilutions) and with extensive washing, sections were mounted with VectaShield and stained with DAPI (Vector Laboratories). Immunofluorescence images were obtained and analyzed using the Nikon Eclipse Ti inverted microscope or by the Zeiss LSM710 confocal microscope.

## Statistics and reproducibility

Quantitative data are presented as mean ± standard error of mean (SEM) from at least 3 independent experiments. When appropriate, statistically differences between multiple comparisons were analyzed using the one-way ANOVA test with Bonferroni correction. Differences were considered significant at $p < 0.05$. All samples, if preserved and properly processed, were included in the analyses, and no samples or animals were excluded, except for mice with conventional surgery damage. No statistical method was used to predetermine sample size, and all experiments except for those involving animals were not randomized. Immunoblotting, reporter assay, and qRT-PCR experiments were repeated to a minimum of 3 independent experiments to ensure reproducibility. The investigators were not blinded to allocation during experiments and outcome assessment.

## Supporting information

**S1 Fig. PPM1A facilitates both nuclear distribution and transcription potency of YAP/ TAZ. (A)** The protein levels of individual phosphatases used in the phosphatome screening was shown by the immunoblotting targeting their Flag tags. **(B)** Transcription potency of TAZ, which was suppressed by coexpression of MST1, was profoundly recovered by cotransfection of wild-type PPM1A but not its phosphatase-dead form (D239N). **(C)** The activity of TEAD-responsive promoter was enhanced in the inducible expression of PPM1A in a dose-dependent manner. **(D)** Depletion of YAP down-regulated the mRNA levels of CYR61 and CTGF, an indication to validate them as target genes of the YAP/TAZ-TEADs complex. Unprocessed images of blots are shown in S1 Raw Images. Statistics source data are provided in S1 Data. MST1, mammalian sterile 20-like kinase 1; PPM1A, protein phosphatase magnesium-dependent 1A; TAZ, transcriptional coactivator with PDZ-binding motif; TEAD, transcriptional enhanced associate domain; YAP, Yes-associated protein.
(TIF)

**S2 Fig. Deletion or depletion of PPM1A enhances Hippo signaling and inactivates YAP/ TAZ. (A)** Glucose starvation and PPM1A KO similarly triggered the up-regulation of phospho-YAP (S127). **(B)** Genetic ablation of PPM1A in HEK293 cells resulted in an enhanced level of phospho-YAP (S127) and an exaggerated TAZ degradation, in a degree similar to cells with energy deficiency. **(C)** Reintroduction of ectopic PPM1A in PPM1A KO HEK293 cells restored the suppressed activity of the YAP/TAZ-TEAD promoter. **(D, E)** The decreased mRNA levels of CTGF (D) and CYR61 (E) were detected by qRT-PCR assays in the absence of

endogenous PPM1A. **(F)** A statistics for cells with the dominant nucleo-YAP in MEFs from WT and PPM1A KO mice. **(G)** An enhanced level of phospho-YAP (S112) was detected in the lysates of livers of young homozygous PPM1A KO mice. **(H)** PPM1A depletion in A549 cells by siRNA interference resulted in an increased phosphorylation level of YAP at the S127 residue. **(I)** Reconstitution of PPM1A restored the nuclear distribution of endogenous YAP/TAZ in PPM1A KO cells. Unprocessed images of blots are shown in S1 Raw Images. Statistics source data are provided in S1 Data. KO, knockout; MEF, mouse embryonic fibroblast; phospho-YAP, phosphorylating forms of YAP; PPM1A, protein phosphatase magnesium-dependent 1A; qRT-PCR, quantitative real-time PCR; siRNA, small interfering RNA; TAZ, transcriptional coactivator with PDZ-binding motif; TEAD, transcriptional enhanced associate domain; WT, wild-type; YAP, Yes-associated protein.
(TIF)

**S3 Fig. PPM1A directly and selectively eliminates YAP phosphorylation. (A)** PPM1A purified from cells was active, as evidenced by its capability to dephosphorylate TBK1, a substrate previously reported [40]. **(B)** Purified PPM1A dephosphorylated YAP, TAZ, and TBK1 during an in vitro phosphatase assay, which required enzymatic activity of PPM1A and $Mg^{2+}/Mn^{2+}$. **(C)** An indication for the successful reconstitution of YAP WT and 2SA mutant in YAP-depleted HCT116 cells. **(D)** Fluorescence intensity analysis by Image J software indicated the obvious presence of cytoplasmic-YAP in PRP4K-expressed cells, which was diminished by PPM1A induction. Unprocessed images of blots are shown in S1 Raw Images. Statistics source data are provided in S1 Data. PPM1A, protein phosphatase magnesium-dependent 1A; TAZ, transcriptional coactivator with PDZ-binding motif; TBK1, TANK-binding kinase 1; WT, wild type; YAP, Yes-associated protein.
(TIF)

**S4 Fig. PPM1A is indispensable for murine intestinal regeneration upon colitis. (A)** Decrease of proliferating cells (Ki67 positive) of $Ppm1a^{-/-}$ intestinal epithelium was calculated. **(B)** Administration of the MST1/2 inhibitor XMU-MP-1 largely prevented DSS-induced disruption of crypts and villus architectures in PPM1A KO mice. **(C, D)** Pharmacological blockade of TGF-β signaling by SB431542, or IFN-I signaling by anti-IFNAR1 neutralizing antibody, failed to protect the DSS-induced colitis. **(E–G)**, Genetic deficiency of LATS1 ($Lats1^{+/-}$) survived PPM1A KO mice from the DSS-induced colitis attack (E), largely preserved the villus structure of $Ppm1a^{-/-}$ intestines (F), and partially recovered the colon length (G). Unprocessed images of blots are shown in S1 Raw Images. Statistics source data are provided in S1 Data. DSS, dextran sulphate sodium; IFN-I, type I interferon; KO, knockout; LATS1, large tumor suppressor kinase 1; MST1/2, mammalian sterile 20-like kinase 1 and 2; PPM1A, protein phosphatase magnesium-dependent 1A; TGF-β, transforming growth factor beta.
(TIF)

**S1 Data. Source data of statistics.**
(XLSX)

**S1 Table. List of recombinant DNA.**
(XLSX)

**S2 Table. Antibodies used in study.**
(XLSX)

**S3 Table. Oligos used in study.**
(XLSX)

**S4 Table. MS results of YAP modification.**
(XLSX)

**S1 Raw Images. Raw images of the WB.**
(PDF)

# Acknowledgments

We are grateful to Drs. Kun-Liang Guan for reagents, Bing Yang for mass spectrometry data analysis, and Junfang Ji for guidance on animal operation technique. Thanks also go to technical assistance by the Biological Mass Spectrometry Analysis Center and Life Sciences Institute core facilities, Zhejiang University.

# Author Contributions

**Conceptualization:** Ruyuan Zhou, Pinglong Xu.

**Data curation:** Ruyuan Zhou, Qirou Wu, Mengqiu Wang, Seema Irani, Pinglong Xu.

**Formal analysis:** Ruyuan Zhou, Qirou Wu, Mengqiu Wang, Seema Irani, Qian Zhang, Yan Jessie Zhang, Pinglong Xu.

**Funding acquisition:** Qian Zhang, Pinglong Xu.

**Investigation:** Ruyuan Zhou, Qirou Wu, Mengqiu Wang, Seema Irani, Xiao Li, Qian Zhang, Fansen Meng, Shengduo Liu, Fei Zhang, Yan Jessie Zhang, Pinglong Xu.

**Methodology:** Ruyuan Zhou, Qirou Wu, Seema Irani, Xiao Li, Fansen Meng, Shengduo Liu, Fei Zhang, Xiaojian Wang, Jian Zou, Hai Song, Jun Qin, Xin-Hua Feng, Yan Jessie Zhang.

**Resources:** Liming Wu, Xia Lin, Xiaojian Wang, Jian Zou, Hai Song, Jun Qin, Tingbo Liang, Xin-Hua Feng.

**Supervision:** Pinglong Xu.

**Validation:** Ruyuan Zhou.

**Writing – original draft:** Pinglong Xu.

**Writing – review & editing:** Xin-Hua Feng, Pinglong Xu.

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
