## [Editor Report · Decision Letter 0]

30 Jun 2020

Dear Dr Xu, 

Thank you for submitting your manuscript entitled "PPM1A dephosphorylates and activates YAP to govern mammalian intestinal and liver regeneration" for consideration as a Research Article by PLOS Biology.

Your manuscript has now been evaluated by the PLOS Biology editorial staff as well as by an academic editor with relevant expertise and I am writing to let you know that we would like to send your submission out for external peer review.

Please re-submit your manuscript within two working days, i.e. by Jul 02 2020 11:59PM.

Kind regards,

Di Jiang, PhD

Senior Editor

PLOS Biology

---

## [Decision Letter · Decision Letter 1]

27 Aug 2020

Dear Dr Xu,

Thank you very much for submitting your manuscript "PPM1A dephosphorylates and activates YAP to govern mammalian intestinal and liver regeneration" for consideration as a Research Article at PLOS Biology. Thank you also for your patience as we completed our editorial process, and please accept my apologies for the delay in providing you with our decision. Your manuscript has been evaluated by the PLOS Biology editors, an Academic Editor with relevant expertise, and by four independent reviewers.

The reviews are attached below. As you will see, the four reviewers generally find this study interesting and potentially worth publishing, however, they identify many points that would need to be addressed in a revised version.

In light of the reviews, we will not be able to accept the current version of the manuscript, but we would welcome re-submission of a revised version that takes into account the reviewers' comments. We cannot make any decision about publication until we have seen the revised manuscript and your response to the reviewers' comments. Your revised manuscript is also likely to be sent for further evaluation by the reviewers.

We expect to receive your revised manuscript within 3 months. 

**IMPORTANT - SUBMITTING YOUR REVISION**

*Re-submission Checklist*

*Published Peer Review*

*PLOS Data Policy*

*Blot and Gel Data Policy*

Sincerely,

Ines

--

Ines Alvarez-Garcia, PhD,

Senior Editor,

ialvarez-garcia@plos.org,

PLOS Biology

Reviewers’ comments

Rev. 1:

In this manuscript, the authors report the identification of PPM1A as a new regulator of YAP function. Zhou et al. use a phosphatome screen to identify putative phosphatases that regulate YAP function and identify PPM1A, a metal-dependent phosphatase as a hit in their screen. PPM1A expression counteracts MST-, LATS- and MAP4K-dependent phosphorylation of YAP at Ser127, one of the critical residues that regulates YAP function and that is involved in the control of YAP nuclear translocation. The authors show that PPM1A expression promotes YAP nuclear localization and correlates with increased YAP target gene expression. The authors also suggest that loss of PPM1A function by CRISPR is associated with increased YAP cytoplasmic retention. PPM1A KO intestinal organoids have proliferation defects and, in vivo, PPM1 KO mice have deficient gut and liver regeneration upon injury. The authors suggest that the in vivo defects of PPM1A KO animals are related to the regulation of YAP and not to the regulation of other pathways where PPM1A has shown to be involved, such as the TGF-beta or IFN pathways.

This report addresses an important question in the Hippo pathway field, regarding the regulation of YAP phosphorylation by phosphatases. So far, it is still unclear whether there are specific phosphatases that directly regulate specific YAP phosphorylation events. However, despite the fact that the manuscript tackles an important point of regulation within the Hippo pathway, there are particular issues with the manuscript that prevent its publication in its current form. Below are points to be addressed by the authors.

Major points:

The main criticism of the manuscript relates to the fact that the authors suggest that PP1MA directly targets YAP and dephosphorylates it. However, in my opinion, the data supporting this conclusion is not robust enough.

The authors fail to address in the manuscript whether the effect of PPM1A on YAP phosphorylation is indeed direct or due to changes in the phosphorylation of upstream components (e.g. MST1). This is an important point to address as it relates to the mechanistic effect of PPM1A in the pathway. Since the in vitro phosphatase assay is not performed with recombinant PPM1A but with PPM1A immunoprecipitated from cells, it is possible that the effect is indirect and not due to PPM1A itself. Although the authors show an in vitro kinase assay where PPM1A supposedly dephosphorylates YAP on Ser127, Fig. 4A lacks a positive control to ensure that PPM1A purified from cells is active. Ideally, a known target of PPM1A should be tested in the same conditions. Moreover, the MS data provided by the authors shows that LATS is detected in the control condition but not when PPM1A is added to the reactions. Why would this be the case? And could this account for the differences in YAP phosphorylation levels? If LATS remained in a complex with YAP in one condition but not the other, this could obviously affect the extent of phosphorylation that occurs and the final levels of p-YAP in the in vitro kinase assay.

Related to this point, data shown in Fig. 2H regarding the levels of p-LATS should be quantified. Does PPM1A affect the levels of p-LATS in Thr1079? If so, is the effect of PPM1A on YAP phosphorylation actually an indirect effect of controlling LATS phosphorylation?

If PPM1A is indeed essential to regulate YAP Ser127 phosphorylation, one would expect that the KO phenotypes would be more severe and that the mice would not be viable. The authors suggest that PPM1B may compensate for PPM1A as the double KO mice were lethal. Do the authors have any data that supports this argument, either in vitro or in vivo? Are the in vitro phenotypes associated with PPM1A loss of function altered when PPM1B levels are modulated?

If the function of PPM1A in the Hippo pathway is indeed to mainly regulate Ser127 phosphorylation, its effects should be dispensable when cells express a non-phosphorylatable version of YAP. This should be tested.

Lines 394-396: The authors state that PPM1A and YAP dephosphorylation are necessary for regeneration. However, there is no data on YAP Ser127 phosphorylation from in vivo experiments.

How was the analysis of the Mass Spectrometry data performed? There is no corresponding Methods section for the MS experiments, no mention of how many times the experiments were performed and whether experiments included technical replicates or not. The data in Suppl Table 1 for Fig. 5 is not clear enough to understand what type of analysis was done. Were the total number of YAP peptides found in the experiments 21 (Control) and 17 (PPM1A) as suggested in Supp Table 5? Can the levels of phosphorylation be compared without a reliable form of quantification? Differences in total protein amount, coverage and peptide detection could have a significant impact in the interpretation of the results. Is there a specific reason why peptides spanning the Ser127 site are not detected in the MS samples?

Is PPM1A specific for YAP or does it also target TAZ? If it is the former, how is PPM1A specifically targeted to YAP?

With regards to the results from the phosphatome screen (Fig. 1A and 1B) why is there such a big difference between YAP+MST1 in Fig. 1A and Fig. 1B? In Fig. 1B, YAP+MST1 leads to virtually no activation of YAP, whereas in Fig. 1A there is a ~200-fold induction compared to negative controls. If possible, results from the phosphatome screen and related figures should have accompanying data regarding the levels of the relevant proteins in the assay. Could the differences in YAP readout activation be due to changes in protein levels?

Line 230-232: Authors suggest that PPM1A localization is controlled by LATS. How does this correlate to the fact that in Fig. 1J, PPM1A levels seem to be equivalent in the nucleus and cytoplasm in basal conditions?

In Fig. 2E, why is there a complete absence of p-YAP Ser127 in the absence of 2-DG? In previous experiments under the same conditions (Fig. 1E) there is extensive YAP phosphorylation.

Data related to Fig. 2F should be quantified. Does reconstitution of the KO MEFs rescue YAP subcellular localization?

Lines 227-229: Based on the data shown in Fig. 3D, it is impossible to determine if there is indeed co-localization of PPM1A and YAP, or PPM1A and LATS in the cytoplasm. The readout used is not sensitive enough to determine whether the proteins are indeed physically interacting in the cytoplasm.

I fail to see a significant difference between controls and PPM1A-expressing cells in Fig. 4F.

Some of the critical experiments should be performed in the absence of Mg2+ to enhance the argument of the authors that it is indeed PPM1A that is involved in the regulation of YAP.

Why and how does PPM1A target several residues in YAP? Is there a specific consensus for PPM1A and do the YAP sites that seem to be dephosphorylated by PPM1A fit that consensus sequence?

The number of animals used in the DSS experiments seems low, considering that this procedure is known to lead to experimental variability.

How does the data on TGF-beta shown in Fig. 7D correlate with the fact that previous studies have shown that TGF-beta can influence regeneration in the liver?

Minor points:

Manuscript should be revised in terms of grammar for enhanced understanding by the readership.

Line 79: Typo in Mats.

Line 111: Recent publications revealing the importance of STRIPAK in the regulation of mammalian Hippo signalling should be included as references.

Line 146: Typo in MAP4K1

Fig. S2E should include PPM1A WB data showing levels of PPM1A in the different genotypes.

Analysis of YAP sub-cellular localization in organoids is not entirely clear.

Fig. 7B: Typo in regeneration. Missing information next to images of livers (presumably statistical analysis?).

Rev. 2:

This manuscript from Zhou et al revealed that PPM1A phosphatase activity positively regulate mammalian liver and intestinal regeneration by targeting its newly identified substrate YAP. The authors employed an "unbiased phosphatome screening" consisting of cDNA of 40 known serine/threonine phosphatases to kick-start their investigation via YAP dependent transcriptional activity as a readout. Along with other reported phosphatases, the author identified PPM1A regulate the phosphorylation status of YAP at crucial Ser127 (as well as Ser109 and Ser366) thereby translocating YAP to the nucleus for YAP-TEAD transcription. In addition, PPM1A can also regulate phosphorylation status of YAP in the nucleus. The authors further provided evidences of PPM1A physical interactions with YAP using confocal microscopy and co-immunoprecipitation. Finally, the authors showed that PPM1A phosphatase activity was indeed physiologically important for YAP-TEAD mediated downstream effects. PPM1A phosphatase activity dictates YAP activation and nuclear localization in both cell and animal models, and absence of PPM1A revealed expected disruption in YAP-TEAD dependent organ regeneration and repair, the impairment could be rescued by inhibiting kinase activity of MST1/2.

It is a well conducted study on the elucidation of involvement and roles of PPM1A in Hippo-YAP signalling. With the questions below addressed, I feel that it is worth publishing in PLOS Biology.

1)In the abstract (line 48-49) and the highlight section (line 73-74), the authors stated that phenotypes of PPM1A KO mice inability in organ regeneration was rescued by prevention of PPM1A-YAP regulation. If I were to understand correctly, only allowing PPM1A to regulate YAP will then allow normal regeneration of injured organ. How can prevention of PPM1A-YAP regulation rescued the regeneration?

2)Can the author comment how may PPM1A be localized (cytosol or nucleus) in basal condition? From the data available in the text, PPM1A seems to follow YAP localization? As the author mentioned in the main text, HA-YAP overexpressed cell model resulted in PPM1A mainly nuclear localization while if HA-LATS1 is overexpressed, PPM1A can exist substantially in the cytosol. Perhaps PPM1A localized based on YAP phosphorylation status? How much a role can PPM1A act as a phospho-YAP sensor?

3) Edits: The author should standardize the annotation of nucleus marker "DAPI" or "DAP1" in the figures.

Rev. 3:

The Hippo pathway is an essential organ size and homeostatic pathway. The authors investigated the mechanism for Yap/Taz dephosphorylation and discovered that PPM1A/PP2Cα as an important p'ase to counter the Hippo mediated phosphorylation of Yap/Taz. Overall the data are convincing and this is an important advance for the Hippo field. There are many areas that can be improved in the writing and also some data presentation:

Comments:

1) English usage should be carefully re-evaluated throughout the manuscript. In some places, such as abstract, the manuscript is confusingly written.

2) This statement regarding Figure 2B should be clarified as it is unclear if it is accurate: "Intriguingly, the extent of YAP phosphorylation in resting state was comparable to

those YAPs under glucose starvation/energy stress."

3) please state the phenotype of the PPM1A mutant mice early in the manuscript rather than at the end. If this is the main p'ase for Yap/Taz it would be predicted to be embryonic lethal

4) Figure 3C is an important data point and should be quantified to make the point more conclusively.

5) For subcellular localization of endogenous PPM1A there should be a control such as siRNA against PPM1A (Fig 3F).

6) The interaction data at end of Figure 3 would be strengthened by looking at endogenous proteins if possible.

7) Data in Fig 4 A, B are convincing and support the hypothesis

8) Data in figure 5D (organoids) needs to be improved - current version is poor quality

9) Data in Fig 6 F can also be improved

Rev. 4:

In this manuscript PBIOLOGY-D-20-01910R1 entitled 'PPM1A dephosphorylates and activates YAP to govern mammalian intestinal and liver regeneration', Zhou et al. presented an impressive amount of biochemical data to prove that protein phosphatase magnesium dependent 1A (PPM1A), a member of PP2C subfamily phosphatases, was a direct modifier of YAP. Accordingly, genetic ablation and depletion of PPM1A resulted in YAP/TAZ cytoplasmic retention, while PPM1A deficiency in organoids and mice resulted in reduced nucleo-YAP and diminished cell proliferation, which leading to severe regeneration defects in gut epithelium during colitis and in livers upon injury. Although these findings were potentially interesting, the authors should address the following concerns.

Main concerns:

1. Control is missing in Figure 1G, 2D, 2K-2L, S2C-S2D, and 3A which the mRNA levels of CTGF and CYR61 in corresponding YAP overexpression or knockdown/knockout are expected to be tested for their expression.

2. How to explain that PPM1A is mainly located in the nucleus while the pYAP is usually retained in the cytoplasm? Figure 1H and 3C, lower bands were supposed to be endogenous p-YAP(s127) However, endogenous p-Yap was not altered by PPM1A overexpression in cells. Please address these inconsistence.

3. The shift Phos-Tag bands have a different trend in PPM1A knockout cells in Figure 2B, why? The author should add LATS1 as control in Figure 3B; Why LATS1 S909D and 1079D caused different changes in pYAP in Figure 3C?

4. In Figure 2J, depletion of PPM1A in HEK293 significantly decreased TEAD-driven activity both in resting state and under serum starvation. However, in Figure 2C, depletion of PPM1A in HEK293 attenuated TEAD-driven activity only in resting state but not serum starvation state, why?

5. Figure 2B, the levels of p-YAP in resting state was comparable to those under glucose starvation/energy stress. However, in previous study, glucose starvation significantly increased YAP phosphorylation. (1.AMPK modulates Hippo pathway activity to regulate energy homeostasis. 2.Cellular energy stress induces AMPK-mediated regulation of YAP and the Hippo pathway). The authors should address these inconsistent findings.

6. If PPM1A is mainly responsible for directly eliminating phospho-S127 on YAP, why does PPM1A interact with LATS1? (Figure 3I)? Are PPM1A and LATS1 competitively combined with pYAP/YAP?

7. Why is XMU-MP-1 not used in figures 5G-H and 7D, but in figure 6? The author should add the phenotype of XMU-MP-1 treatment in the corresponding experiments.

8. The lower panel in Figure 7B needs to be replaced with a cleaner one.

---

## [Decision Letter · Decision Letter 2]

8 Jan 2021

Dear Dr Xu,

Thank you for submitting your revised Research Article entitled "PPM1A dephosphorylates and activates YAP to govern mammalian intestinal and liver regeneration" for publication in PLOS Biology. I have now obtained advice from two of the original reviewers and have discussed their comments with the Academic Editor. 

Based on the reviews, we will probably accept this manuscript for publication, assuming that you will modify the manuscript to address the data and other policy-related requests noted at the end of this email. We would also like you to consider a suggestion to improve the title making a change as follows:

"The protein phosphatase PPM1A dephosphorylates and activates the transcription factor YAP to govern mammalian intestinal and liver regeneration"

We expect to receive your revised manuscript within two weeks. Your revisions should address the specific points made by each reviewer. 

-  a cover letter that should detail your responses to any editorial requests. 

*Published Peer Review History*

*Early Version*

Sincerely,

Ines

--

Ines Alvarez-Garcia, PhD,

Senior Editor,

PLOS Biology

ETHICS STATEMENT:

-- Thank you for providing the ethics statement, but please include also the license or approval number.

Fig. 1A-D, H, I; Fig. 2A, C, D, G, J-L; Fig. 3A, B, F; Fig. 4C, E-H; Fig. 5D, G, H, I-K; Fig. 6A-C, F, H; Fig. 7A-E, G; Fig. S1B-D; Fig. S2C-F; Fig. S4C, D (please renumber as Fig. S3) and Fig. S6A, C-E (please renumber as Fig. S4)

Note that all the Supplementary files have to be numbered independently of the main figures, so please renumber them from 1-4.

The numerical data provided should include all replicates AND the way in which the plotted mean and errors were derived (it should not present only the mean/average values).

BLURB

Please provide a blurb which (if accepted) will be included in our weekly and monthly Electronic Table of Contents, sent out to readers of PLOS Biology, and may be used to promote your article in social media. The blurb should be about 30-40 words long and is subject to editorial changes. It should, without exaggeration, entice people to read your manuscript. It should not be redundant with the title and should not contain acronyms or abbreviations. For examples, view our author guidelines: https://journals.plos.org/plosbiology/s/revising-your-manuscript#loc-blurb

Reviewers’ comments

Rev. 1:

The revised version of the manuscript by Zhou et al. is a markedly improved report that more convincingly shows that PPM1A is involved in the regulation of YAP. Importantly, the authors addressed appropriately the major points that required improvement before the manuscript could be recommended for publication in PLoS Biol.

In particular, the fact that the authors now show that recombinant PPM1A can dephosphorylate YAP in vitro suggests that, indeed, PPM1A is acting directly on YAP. Moreover, as suggested by the reviewers, the authors also provide evidence that, as predicted by their model and the proposed mechanism of action of PPM1A, a non-phosphorylatable form of YAP is largely refractory to the action of PPM1A. The authors also provide evidence that supports their claim that PPM1B could compensate for PPM1A loss in the regulation of YAP. Although this is not entirely resolved given the inability of testing this adequately in vivo, the in vitro results showing an enhancement of YAP phosphorylation when both phosphatases are depleted, suggests this compensatory mechanism is possible. The fact that the in vivo experiments have been complemented by the use of a larger number of animals also gives strength to the authors' claims that PPM1A is involved in regeneration in vivo.

In conclusion, the authors have provided a sufficiently enhanced manuscript, which is suitable for publication.

Rev. 4:

My concerns have been addressed satisfactorily.

---

## [Editor Report · Decision Letter 3]

26 Jan 2021

Dear Dr Xu,

Thank you for submitting your revised Research Article entitled "The protein phosphatase PPM1A dephosphorylates and activates YAP to govern mammalian intestinal and liver regeneration" for publication in PLOS Biology.

We have now checked the remaining editorial requests and we are mostly satisfied, but there is one very minor issue pending:

- Please indicate in the supplementary figure legends where the data can be found.

We expect to receive your revised manuscript within one week.

Sincerely,

Ines

--

Ines Alvarez-Garcia, PhD,

Senior Editor,

PLOS Biology

---

## [Editor Report · Decision Letter 4]

29 Jan 2021

Dear Dr Xu,

On behalf of my colleagues and the Academic Editor, Nic Tapon, I am pleased to say that we can in principle offer to publish your Research Article "The protein phosphatase PPM1A dephosphorylates and activates YAP to govern mammalian intestinal and liver regeneration" in PLOS Biology, provided you address any remaining formatting and reporting issues. These will be detailed in an email that will follow this letter and that you will usually receive within 2-3 business days, during which time no action is required from you. Please note that we will not be able to formally accept your manuscript and schedule it for publication until you have made the required changes.

PRESS

Thank you again for supporting Open Access publishing. We look forward to publishing your paper in PLOS Biology. 

Sincerely, 

Ines

--

Ines Alvarez-Garcia, PhD 

Senior Editor 

PLOS Biology
